# Pervasive associations between dark septate endophytic fungi with tree root and soil microbiomes across Europe

Tarquin Netherway [1] ✉, Jan Bengtsson [1], Franz Buegger[2], Joachim Fritscher[3,4], Jane Oja[5], Karin Pritsch [2], Falk Hildebrand[3,4], Eveline J. Krab [6] & Mohammad Bahram [1,5]

Trees interact with a multitude of microbes through their roots and root symbionts such as mycorrhizal fungi and root endophytes. Here, we explore the role of fungal root symbionts as predictors of the soil and root-associated microbiomes of widespread broad-leaved trees across a European latitudinal gradient. Our results suggest that, alongside factors such as climate, soil, and vegetation properties, root colonization by ectomycorrhizal, arbuscular mycorrhizal, and dark septate endophytic fungi also shapes tree-associated microbiomes. Notably, the structure of root and soil microbiomes across our sites is more strongly and consistently associated with dark septate endophyte colonization than with mycorrhizal colonization and many abiotic factors. Root colonization by dark septate endophytes also has a consistent negative association with the relative abundance and diversity of nutrient cycling genes. Our study not only indicates that root-symbiotic interactions are an important factor structuring soil communities and functions in forest ecosystems, but also that the hitherto less studied dark septate endophytes are likely to be central players in these interactions.

Symbioses between plants and fungi are the most widespread and integrated biotic interactions in terrestrial ecosystems, with their distributions largely driven by climate and edaphic gradients[1-3]. Most trees form symbioses with ectomycorrhizal (EcM) or arbuscular mycorrhizal (AM) fungi and other symbionts such as root endophytic fungi, which extensively colonize root systems[4] and promote the functioning of their hosts[5,6]. Directly and via these symbiotic fungi, trees interact with numerous other soil and root-associated microbes, mainly bacteria and fungi, which are important for carbon (C) and nutrient cycling, as well as plant health and growth[7-9]. Growing evidence suggests that either directly via competition with other microbes or indirectly by influencing soil processes[10], EcM and AM

fungi affect soil microbial communities and their functions in different ways and contribute to the proposed conservative organic versus acquisitive inorganic nutrient economies of their hosts, respectively[10-13]. Therefore, the dominant tree mycorrhizal type has often been used as a proxy for soil microbiome functioning. For example, tree mycorrhizal type has been used to explain the activity of soil saprotrophs[14-16], the prevalence of plant pathogens[13,17], fungal diversity[18,19], the bacteria/fungi biomass ratio[13,20,21], and the activity of enzymes involved in nitrogen (N), phosphorus (P), and C acquisition[21]. Yet, an issue with the binary assignment of a mycorrhizal type (EcM or AM) to a tree or forest is that it ignores the actual presence or prevalence of the root symbioses and complex root associations such as

[1]Department of Ecology, Swedish University of Agricultural Sciences, Ulls väg 16, 756 51, Uppsala, Sweden. [2]Research Unit for Environmental Simulation (EUS), German Research Center for Environmental Health, Helmholtz Zentrum München, Ingolstaedter Landstr. 1, 85764 Neuherberg, Germany. [3]Quadram Institute Bioscience, Norwich Research Park, Norwich, Norfolk NR4 7UQ, UK. [4]Digital Biology, Earlham Institute, Norwich Research Park, Norwich, Norfolk NR4 7UQ, UK. [5]Department of Botany, Institute of Ecology and Earth Sciences, University of Tartu, 40 Lai St, Tartu, Estonia. [6]Department of Soil and Environment, Swedish University of Agricultural Sciences, Lennart Hjelms väg 9, 750 07 Uppsala, Sweden. ✉e-mail: tarquin.netherway@slu.se

dual-mycorrhiza[10,22,23]. Furthermore, little attention has been paid to the potential role of other common co-occurring and widespread tree root-associated fungi, such as dark septate endophytes (DSE)[24], which may be just as important as mycorrhizal associations in shaping belowground communities and functions.

The ecological function of the facultative, host generalist, common, prolific, and root colonizing DSE, and root endophytes in general, is still under consideration, and they probably act on a spectrum between free-living saprotrophs, mycorrhizal fungi, and parasites[25,26]. While they are taxonomically poorly defined, DSE fungi are morphologically characterized by the presence of melanized septate (ascomycetous) hyphae and occasionally microsclerotia in living plant roots[26]. Increasing evidence from agricultural systems shows that these enigmatic fungi can also be associated with enhanced plant performance and stress tolerance under harsh environmental conditions by enhancing access to nutrients and protecting against pathogens[27,28]. While DSE have largely been overlooked in forests in favor of mycorrhizal associations, they may be just as abundant, if not more so, in forest ecosystems when considering the number of host plants they colonize and the extent to which they colonize root systems[25]. Thus, assessing the influence of multiple tree-fungal associations on soil and root microbiomes across large environmental gradients will help us identify the main biotic and abiotic determinants of the structure and function of microbial communities to better understand forest ecosystem processes.

We performed a large-scale field study on the potential role of root colonization by EcM, AM, and DSE fungi in shaping soil and root microbiomes. We sampled the soil and roots of 305 forest trees from 3 different deciduous broadleaf genera (*Alnus*: dual mycorrhizal/N-fixing *Frankia* bacteria, *Betula*: EcM, and *Sorbus:* AM). We considered climate, soil, and vegetation properties across 18 sites over a 3200 km European latitudinal gradient (Fig. 1a & b; Supplementary Data 1). A highly consistent sampling design and processing were used, and to analyze soil and root microbial communities, we utilized metabarcoding alongside shotgun metagenomics for the analysis of functional genes and estimated litter decomposition using tea bags. We compared these properties to the root colonization rates of EcM, AM, and DSE fungi on widespread trees. We hypothesized that both mycorrhizal and DSE fungal colonization would affect soil and root microbiome structure and potential functions in addition to and mediated by climate and soil properties. More specifically, we expected that, as they often monopolize roots, colonization by mycorrhizal and DSE fungi would suppress plant pathogens while also altering the relative abundances of taxa and genes involved in decomposition and nutrient cycling. We show that DSE colonization is more strongly and consistently associated with the structure and potential function of tree root and soil microbiomes across our sites than mycorrhizal colonization and many abiotic factors, indicating that DSE fungi are potentially important mediators of plant-soil interactions.

## Results
### Root symbiont colonization and their relative abundances in soil and roots
Root colonization rates (determined based on microscopic quantification) and the relative abundance of root symbionts (based on centered-log ratio transformed metabarcoding reads) were best explained by moisture availability for EcM fungi, soil pH (colonization) and host tree basal area (relative abundance) for AM fungi, soil carbon/nitrogen (C/N) (colonization) and pH (relative abundance) for DSE fungi, and soil pH for N-fixing *Frankia* bacteria (relative abundance). The average root colonization by EcM fungi varied between 40 and 85% across the gradient of sites (Fig. 2a). It was most strongly and positively correlated with soil moisture ($R^2$m = 0.36, p = 1e-06) (Fig. 2a) and was generally poorly explained by factors other than climatic

moisture deficit (CMD) (Fig. S1). The relative abundance of EcM fungi on roots was best explained by a negative association with CMD ($R^2$m = 0.36, p = 1e-05) (Fig. 2b) It was also positively correlated with soil C/N, soil moisture, and the coniferous EcM tree basal area, while it was negatively correlated with mean annual temperature (MAT) (Fig. S1; Supplementary Note 1). The relative abundance of EcM fungi on roots was positively correlated with EcM colonization ($R^2$m = 0.09, p = 0.019) (Fig. 2c) and the relative abundance of EcM fungi in soil, which also showed similar associations with environmental factors (Fig. S1 and Supplementary Note 1).

Compared to EcM colonization, AM root colonization was low, sporadic, and absent on many sampled trees, with averages varying between 0 and 25% (Fig. 2d). Out of environmental factors, AM colonization was positively associated with soil pH ($R^2$m = 0.14, p = 0.011) (Fig. 2d). The relative abundance of AM fungi in roots was best explained by a negative association with the basal area of EcM/AM trees ($R^2$m = 0.24, p = 0.002) (Fig. 2e); the same relationship was observed for AM fungi in soil (Fig. S1). The relative abundance of AM fungi in roots was positively correlated with AM colonization ($R^2$m = 0.18, p = 0.003) (Fig. 2f), as was the relative abundance of AM fungi in soil with AM fungi in roots, AM fungi in soil, and AM colonization (Fig. S1; Supplementary Note 1).

Root colonization by DSE was generally more variable compared to AM and EcM colonization, with averages ranging between 0 and 50%. It was best explained by a positive association with soil C/N ($R^2$m = 0.39, p = 3e-05) (Fig. 2g), correlated positively with the basal area of coniferous EcM trees and negatively with MAT (Fig. S1; Supplementary Note 1). The relative abundance of potential DSE fungi in roots (see Supplementary Data 2 for a list of fungi considered as potential DSE) was best explained by a negative relationship with soil pH ($R^2$m = 0.32, p = 3e-05) (Fig. 2h) and a positive correlation with DSE colonization ($R^2$m = 0.20, p = 0.002) (Fig. 2i). It was also positively correlated with soil C/N and the coniferous EcM tree basal area (Fig. S1; Supplementary Note 1). Given the ambiguity of assigning fungi as DSE based on metabarcoding, we also considered the relationship between DSE colonization and the relative abundance of fungi in roots with a primary lifestyle classification as root endophytes as well all fungal genera with a potential root endophytic capacity according to the FungalTraits tool[29] and we did not find a significant relationship for either of these classifications (Fig. S2).

While we did not quantify N-fixing nodules formed by bacteria from the genus *Frankia* on *Alnus* roots, the relative abundance of *Frankia* on *Alnus* roots was best explained by soil pH out of environmental factors (Fig. S2). The relative abundance of *Frankia* on roots was also positively associated with the relative abundance of N fixing genes in roots, the diversity of N cycling genes in soil and roots, the diversity of total bacterial functional gene diversity in roots, and the ratio of bacteria/fungi in soil (based on metagenomic read abundances; see Methods) (Fig. S2).

### Root colonization by DSE had a negative relationship with the relative abundance of plant pathogens and a positive relationship with soil saprotrophs
There was a significant negative relationship between DSE colonization and the relative abundance of putative plant pathogens in soil ($R^2$m = 0.29, p = 9e-04) (Fig. 3a) and roots ($R^2$m = 0.10, p = 0.036) (Fig. S3). Structural equation modelling (SEM) showed that DSE colonization had a strong direct negative association (path) with the relative abundance of plant pathogens in soil (standard coefficient −0.44, p = 0.003) (Fig. 4a) after accounting for other explanatory variables such as soil properties and climate in the best fitting model. Root colonization by DSE colonization also had a significant (positive) correlation with the relative abundance of soil saprotrophs (Fig. S3); however, we could not separate this effect from soil C/N and pH with a non-significant SEM path. Neither EcM nor AM colonization correlated with

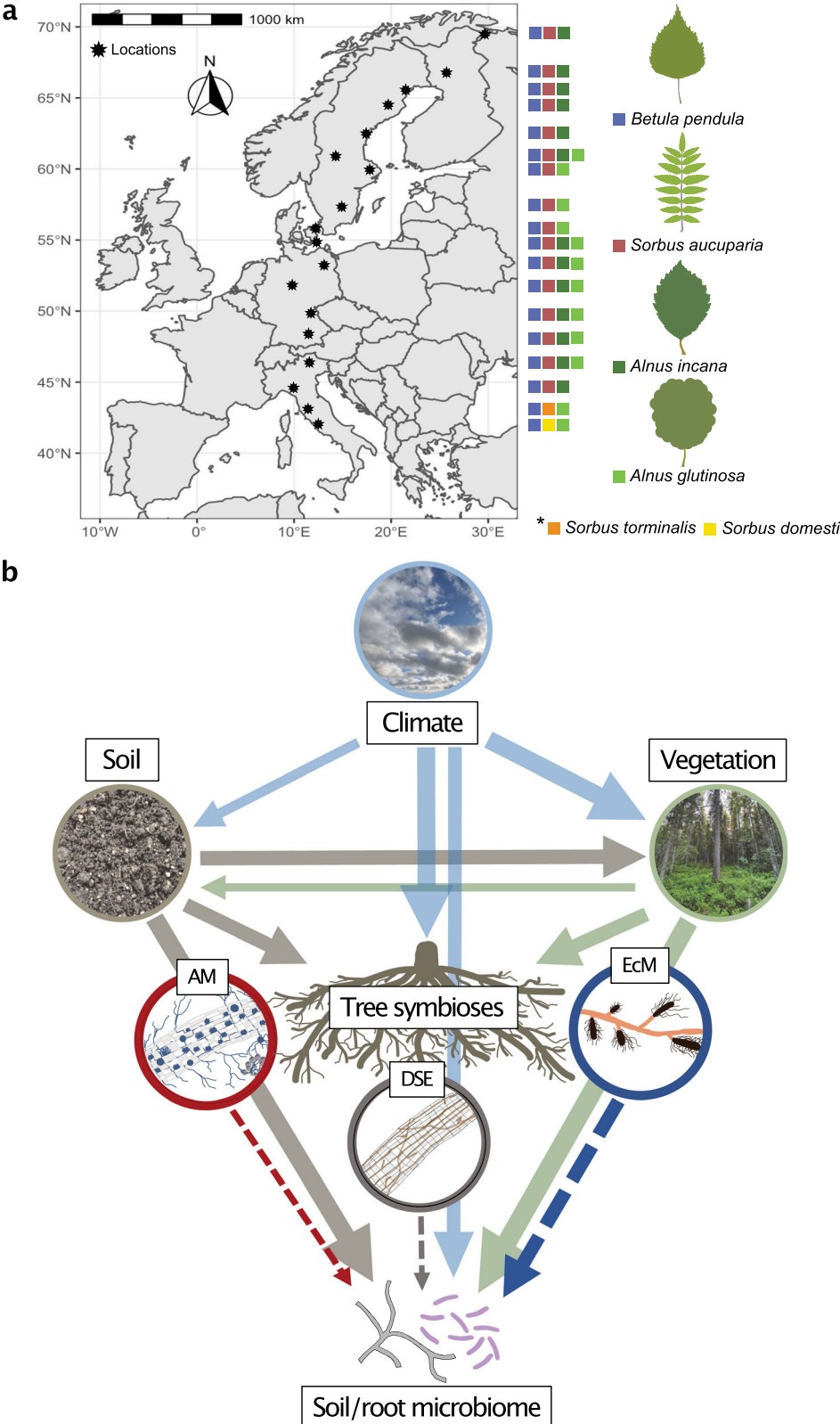

**Fig. 1 | Exploring the associations between root symbioses and soil and root microbiomes across Europe. a** the sampling locations and presence of studied tree species across a European latitudinal gradient from northern Norway to central Italy, and (**b**) the ecological context of the study presented as an a priori path diagram of relationship structures between climatic variables (light blue arrows), vegetation community properties (green arrows), soil properties (brown arrows), tree root symbioses including arbuscular mycorrhizal (AM: dashed red arrow), dark septate endophyte (DSE: dashed gray arrow), and ectomycorrhizal (EcM: dashed dark blue arrows) symbioses, with soil and root microbiome properties as response variables. Direction of arrows indicate the direction of proposed relationships and thickness of arrows indicates the strength of proposed relationships. This figure is adapted from[136] under CC BY 4.0.

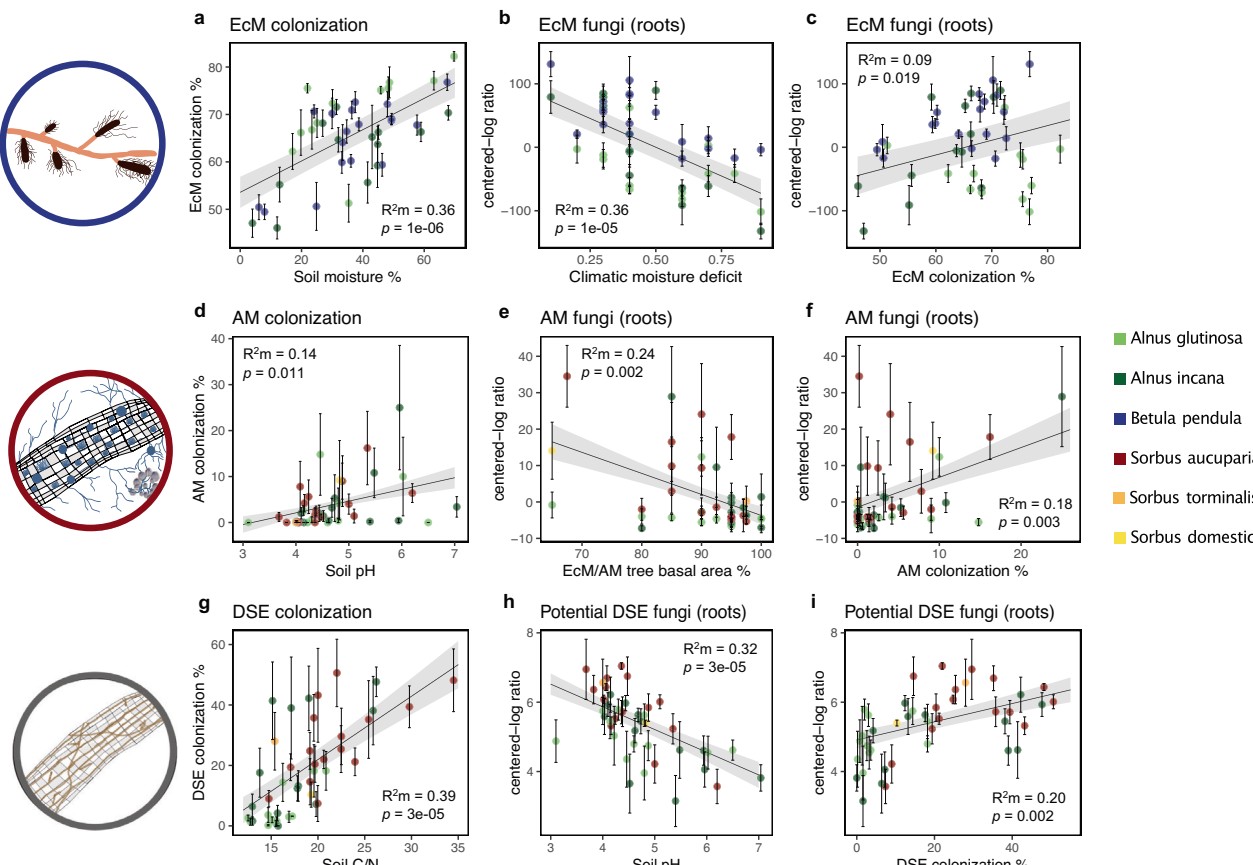

**Fig. 2 | Different tree symbioses respond to contrasting biotic and abiotic factors.** Results of the best fitting linear mixed-effects models showing (**a**) ecto-mycorrhizal (EcM) colonization (% root tips colonized) as explained by gravimetric soil moisture content, ($n = 43$), (**b**) the relative abundance of EcM fungi (centered-log ratio of metabarcoding reads) on the roots as explained by climatic moisture deficit ($n = 43$), (**c**) the relative abundance of EcM fungi on roots as explained by EcM colonization ($n = 43$), (**d**) arbuscular mycorrhizal (AM) colonization (% root length colonized) as explained by soil pH ($n = 43$), (**e**) the relative abundance of AM fungi on roots (centered-log ratio of metabarcoding reads) as explained by the relative basal area of EcM/AM trees ($n = 43$), (**f**) the relative abundance of AM fungi on roots as explained by AM colonization ($n = 43$), (**g**) dark septate endophyte (DSE) colonization (% root length colonized) as explained by soil carbon/nitrogen (C/N) content ($n = 43$), (**h**) the relative abundance of potential DSE fungi (centered-log ratio of metabarcoding reads) on roots as explained by soil pH ($n = 43$), and (**i**) the relative abundance of potential DSE fungi on roots as explained by DSE colonization ($n = 43$). Colors represent different tree hosts: *Betula pendula* (blue), *Sorbus aucuparia* (red), *S. torminalis* (orange), *S. domestica* (yellow), *Alnus glutinosa* (light green), and *A.incana* (dark green). Data points ($n$) presented are mean values of individual tree samples (>10 m apart) from biologically independent tree species at each independent site (i.e., $n$ = tree species × site), and error bars represent ± the standard error (SE) of the mean. The marginal $R^2$ ($R^2$m) of the fixed effect and $p$ values (calculated using the Satterthwaite approximation) for each linear mixed-effects model (plot embedded in site crossed with tree species as random effects) are listed, and the standard error of the fitted line is shaded gray. The statistical test used was two-sided. For linear-mixed effects model summaries supporting this figure see Supplementary Data 10.

the relative abundance of plant pathogens or soil saprotrophs (Figs. 3b, c and S3).

## Root colonization by DSE had a negative association with the ratio of bacteria/fungi and was negatively correlated with bacterial diversity

Dark septate endophyte colonization had a negative association with the ratio of bacteria/fungi in soil ($R^2$m = 0.18, $p$ = 0.002) (Fig. 3d) (SEM: standard coefficient −0.21, $p$ = 0.035) (Fig. 4b) and the bacteria/fungi ratio in roots ($R^2$m = 0.13, $p$ = 0.015) (Fig. S3). Colonization by DSE was also negatively correlated with bacterial diversity in both soil ($R^2$m = 0.14, $p$ = 0.003) and roots ($R^2$m = 0.09, $p$ = 0.044) (Fig. S3). Neither EcM nor AM colonization correlated with the bacteria/fungi ratio or their diversity in soil or roots (Fig. 3e, f and S3).

## Root colonization by DSE influenced the diversity of different functional genes

Root colonization by DSE had a positive association with the diversity of bacterial carbohydrate-active enzyme (CAZyme) genes in soil ($R^2$m = 0.21, $p$ = 7e-04) (Fig. 3g) (SEM: standard coefficient 0.44, $p$ =

0.003) (Fig. 4c), while it had a negative association with both the diversity of total bacterial functional genes in roots ($R^2$m = 0.36, $p$ = 2e-05) (Fig. 3j) (SEM: standard coefficient -0.58, $p$ = 1e-04) (Fig. 4d) and the diversity of N cycling genes in roots ($R^2$m = 0.37, $p$ = 6e-05) (Fig. 3k) (SEM: standard coefficient -0.40, $p$ = 0.003) (Fig. 4e). In addition, DSE colonization correlated positively with the diversity of total fungal functional and CAZyme genes in soil and negatively with bacterial CAZymes in roots (Fig. S3). The negative association between DSE colonization and the diversity of N cycling genes in roots appeared to be robust against the potential confounding positive effect of N-fixing *Frankia* bacteria, where DSE colonization was a more parsimonious predictor ($R^2$m = 0.37, $p$ = 6e-05, AIC = −37.17) compared to the relative abundance of *Frankia* on roots ($R^2$m = 0.24, $p$ = 0.001, AIC = −35.22). Additionally, DSE colonization continued to have a significant path and a stronger direct association (standardized coefficient) when both variables were included within the same SEM: DSE colonization (standard coefficient −0.37, $p$ = 0.002), compared to the relative abundance of *Frankia* on roots (standard coefficient 0.27, $p$ = 0.006) (for detailed SEM model results, see Supplementary Data 3). Our analysis also revealed no significant relationship between DSE colonization and the

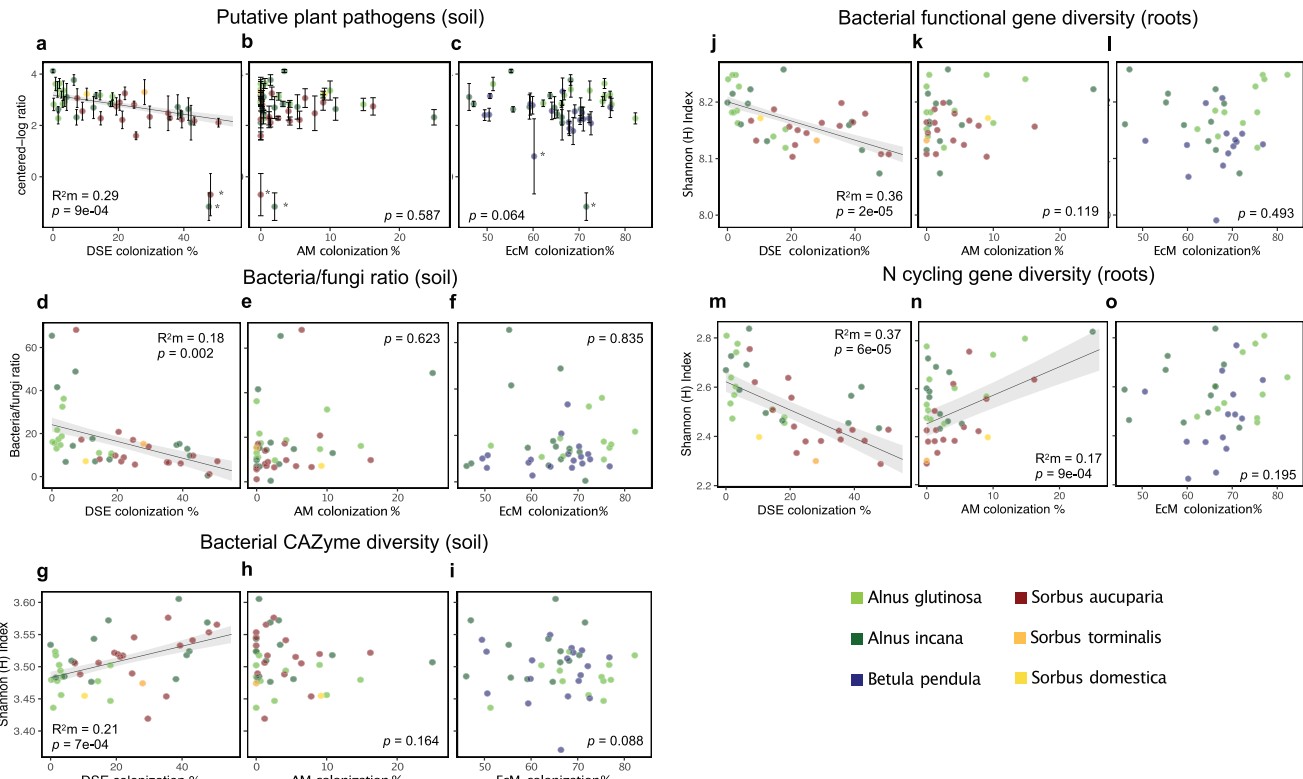

**Fig. 3 | Root symbioses influence the relative abundance of plant pathogens, the bacterial/fungal ratio, and the diversity of microbial functional genes.** **a** dark septate endophyte (DSE) colonization, (**b**) arbuscular mycorrhizal (AM) colonization, and (**c**) ectomycorrhizal (EcM) colonization explaining the relative abundance (centered-log ratio of metabarcoding reads) of putative fungal plant pathogens in soil (the two outliers denoted with an * were excluded from analysis [$n = 41$]); (**d**) DSE colonization, (**e**) AM colonization, and (**f**) EcM colonization explaining the ratio of bacteria/fungi in soil ($n = 43$); (**g**) DSE colonization, (**h**) AM colonization, and (**i**) EcM colonization explaining the diversity (Shannon H Index) of bacterial carbohydrate-active enzymes (CAZymes) in soil ($n = 43$); (**j**) DSE colonization, (**k**) AM colonization, and (**l**) EcM colonization explaining the diversity (Shannon H Index) of total bacterial functional genes in roots ($n = 39$ for DSE and AM colonization; $n = 36$ for EcM colonization); (**m**) DSE colonization, (**n**) AM colonization, and (**o**) EcM colonization explaining the diversity (Shannon H Index)

of nitrogen (N) cycling genes in roots ($n = 39$ for DSE and AM colonization; $n = 36$ for EcM colonization). Colors represent different tree hosts: *Betula pendula* (blue), *Sorbus aucuparia* (red), *S. torminalis* (orange), *S. domestica* (yellow), *Alnus glutinosa* (light green), and *A. incana* (dark green). Data points ($n$) presented in **a**–**c** are mean values of individual tree samples (>10 m apart) from biologically independent tree species at each independent site, and error bars represent ±the standard error of the mean. Data points ($n$) presented in **d**–**o** are composite values of pooled individual tree samples (>10 m apart) from biologically independent tree species at each independent site (i.e., $n$ = tree species × site). The marginal $R^2$ ($R^2$m) of the fixed effect and $p$ values (calculated using the Satterthwaite approximation) from linear mixed-effects models (with plot embedded in site crossed with tree species as random effects) are listed, and the standard error of the fitted line is shaded gray. The statistical test used was two-sided. For linear-mixed effects model summaries supporting this figure see Supplementary Data 11.

relative abundance of *Frankia* on roots ($p = 0.919$) (see all linear-mixed effects model summaries in Supplementary Data 10–20).

By contrast, EcM colonization only had a positive correlation with the diversity of N cycling genes in soil (Fig. S3), and AM colonization only had a positive correlation with the diversity of N cycling genes in roots ($R^2$m = 0.17, $p = 9e-04$) (Fig. 3n) and soil (Fig. S3); however, the EcM and AM relationships did not have significant SEM paths when accounting for other biotic and abiotic factors.

**Alongside climate, soil, and vegetation properties, DSE and EcM colonization influenced the composition of microbial communities and functional genes**

Next, we related AM, EcM, and DSE colonization to the composition of bacteria, fungi, and their functional genes while considering the effect of soil, climatic, and vegetation properties using permutational multivariate analysis of variance (PERMANOVA). Most variance in fungal and bacterial communities and gene compositions was generally attributable to soil, climate, vegetation properties, and the random effect of site (Fig. 5 and S4), although both DSE and EcM colonization were also important for explaining bacterial and fungal community compositions and were particularly important for

explaining the composition of different functional genes in roots (Fig. 5).

Specifically, DSE colonization explained a significant fraction of variation in fungal communities in soil (1.86% explained variance [EV], pseudo-$F = 1.86$, $p = 0.006$) (Fig. 5a) and roots (2.09% EV, pseudo-$F = 1.91$, $p = 1e-04$) (Fig. 5b), bacterial communities in soil (1.30% EV, pseudo-$F = 1.72$, $p = 0.004$) (Fig. 5c) and roots (0.98% EV, pseudo-$F = 1.44$, $p = 0.041$) (Fig. 5d), total fungal functional genes in soil (7.54% EV, pseudo-$F = 5.25$, $p = 1e-04$) (Fig. 5e), total bacterial functional genes in soil (2.28% EV, pseudo-$F = 2.80$, $p = 0.004$) (Fig. 5g) and roots (14.26% EV, pseudo-$F = 9.02$, $p = 4e-04$) (Fig. 5h), N cycling genes in soil (4.19% EV, pseudo-$F = 4.68$, $p = 0.001$) (Fig. 5i) and roots (15.89% EV, pseudo-$F = 10.54$, $p = 1e-04$) (Fig. 5j), P cycling genes in roots (15.99% EV, pseudo-$F = 10.69$, $p = 2e-04$) (Fig. 5k), bacterial CAZymes in roots (16.38% EV, pseudo-$F = 11.28$, $p = 2e-04$) (Fig. 5l), and fungal CAZymes in soil (4.22% EV, pseudo-$F = 3.34$, $p = 0.011$) (Fig. 5m). In addition, EcM colonization explained a significant fraction of variance in the composition of root fungal communities (1.10% EV, pseudo-$F = 1.45$, $p = 0.017$) (Fig. 5b), bacterial communities in soil (1.74% EV, pseudo-$F = 1.89$, $p = 8e-04$) (Fig. 5c) and roots (1.25% EV, pseudo-$F = 1.52$, $p = 0.024$) (Fig. 5d), total fungal functional genes in roots (7.84% EV,

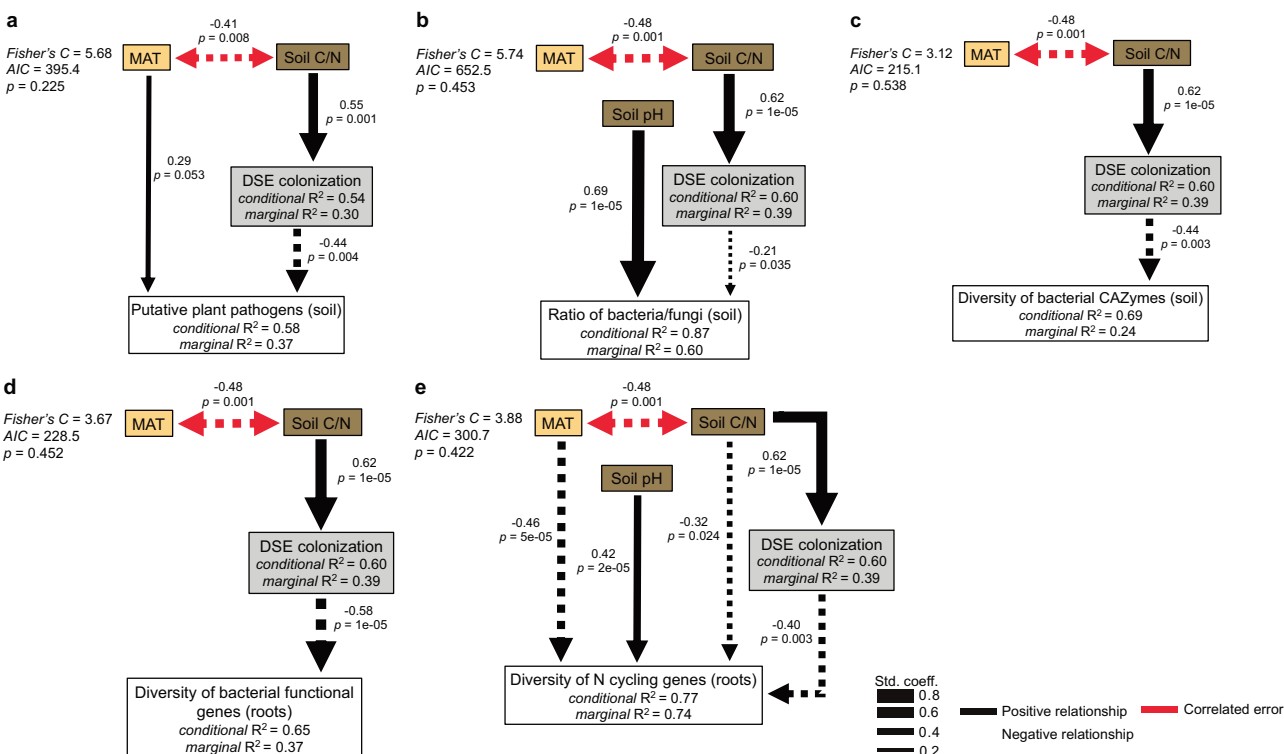

**Fig. 4 | dark septate endophyte colonization has a direct association with the relative abundance of plant pathogens, the bacterial/fungal ratio, and the diversity of different functional genes. a** Structural equation modeling (SEM) of the proposed direct and indirect drivers of the relative abundance (centered-log ratio of metabarcoding reads) of putative fungal plant pathogens in soil ($n = 41$), (**b**) SEM of the proposed direct and indirect drivers of the ratio (metagenomic reads) of bacteria/fungi in soil ($n = 43$), (**c**) SEM of the proposed direct and indirect drivers of the diversity (Shannon H Index) of bacterial carbohydrate-active enzymes (CAZymes) in soil ($n = 43$), (**d**) SEM of the proposed direct and indirect drivers of the diversity (Shannon H Index) of total bacterial functional genes in roots ($n = 39$), (**e**) SEM of the proposed direct and indirect drivers of the diversity (Shannon H Index) of nitrogen (N) cycling genes in roots ($n = 39$). All SEM models were calculated only on data from *Sorbus* and *Alnus* trees on which we measured dark septate endophyte (DSE) colonization ($n =$ tree species × site), which is indicated by gray boxes. Included environmental variables were mean annual temperature (MAT) indicated by yellow boxes, the soil carbon/nitrogen (C/N) ratio, and soil pH, both indicated by brown boxes. All linear mixed-effects models within the SEMs used plot embedded in site crossed with tree species as a random effects structure, only significant ($p < 0.05$) and marginally significant ($p = 0.05$) paths are displayed (calculated using the Kenward-Roger approximation), pseudo $R^2$ values (marginal and conditional) are listed for response variables, the standardized regression coefficients (Std. coeff. listed in the path diagrams above $p$ values), black solid paths indicate a positive relationship, black dashed paths indicate a negative relationship, red paths indicate correlated error. Overall model fit was assessed based on The Akaike information criterion (AIC) and Fisher's C values were used to calculate model $p$ values, with $p > 0.05$ indicating acceptable model fitness. The statistical test used was two-sided. For detailed SEM results see Supplementary Data 3.

pseudo-$F = 3.81$, $p = 0.003$) (Fig. 5f), total bacterial functional genes in soil (1.74% EV, pseudo-$F = 2.30$, $p = 0.018$) (Fig. 5g) and roots (3.32% EV, pseudo-$F = 2.82$, $p = 0.047$) (Fig. 5h), N cycling genes in soil (1.88% EV, pseudo-$F = 2.54$, $p = 0.041$) (Fig. 5i), and fungal CAZymes in roots (10.41% EV, pseudo-$F = 4.82$, $p = 0.006$) (Fig. 5n). Root colonization by AM fungi did not explain any significant fractions (for detailed PER-MANOVA results, see Supplementary Data 4).

### Both root symbiont colonization and soil properties explained more variance in bacterial and fungal communities at mid-latitudes

To examine the mediating effects of climate and geography on the relationships between root symbioses, soil properties, and micro-biomes, we conducted within-site partitioning of variance for bacterial and fungal community compositions in soil and roots across latitude. The combined explained variance by root symbioses (AM, EcM, and DSE colonization) consistently showed unimodal relationships with latitude, indicating enhanced effects in the middle of our latitudinal gradient on soil fungi (Adj $R^2 = 0.18$, $p = 0.047$) (Fig. 6a), root fungi (Adj $R^2 = 0.18$, $p = 0.045$) (Fig. 6b), soil bacteria (Adj $R^2 = 0.25$, $p = 0.020$) (Fig. 6c), and root bacteria (Adj $R^2 = 0.18$, $p = 0.023$) (Fig. 6d). In comparison, the combined explained variance by soil properties (soil pH, C/N, and moisture) had a marginally significant negative relationship with latitude

for soil fungal communities (Adj $R^2 = 0.16$, $p = 0.056$) (Fig. 6a). Similarly, to root symbioses, unimodal relationships with latitude were found for the variance explained by soil properties in the composition of root fungal communities (Adj $R^2 = 0.25$, $p = 0.020$) (Fig. 6b) and root bacterial communities (Adj $R^2 = 0.36$, $p = 0.005$) (Fig. 6d).

### Root colonization by DSE had a consistent association with the relative abundance of CAZymes and nutrient cycling genes

The relative abundance of specific classes of bacterial CAZymes showed associations with all three types of symbiont colonization in soil but only with DSE colonization in roots (Fig. S5 and Supplementary Note 2). The stronger associations in soil were a negative relationship between EcM colonization and bacterial CAZymes targeting cellulose ($R^2$m $= 0.22$, $p = 0.001$), a negative relationship between AM colonization and bacterial CAZymes targeting fungal glucans ($R^2$m $= 0.25$, $p = 0.001$) in contrast to a positive relationship with DSE colonization ($R^2$m $= 0.30$, $p = 2e{-}05$), and a negative relationship between DSE colonization and bacterial CAZymes targeting lignin ($R^2$m $= 0.18$, $p = 0.004$). In roots, DSE colonization was negatively correlated with the relative abundance of all bacterial CAZyme groups (Fig. S5).

For the relative abundance of groups of N cycling genes involved in different processes, DSE colonization was consistently negatively associated with those in soil and roots (Fig. S6; Supplementary Note 3).

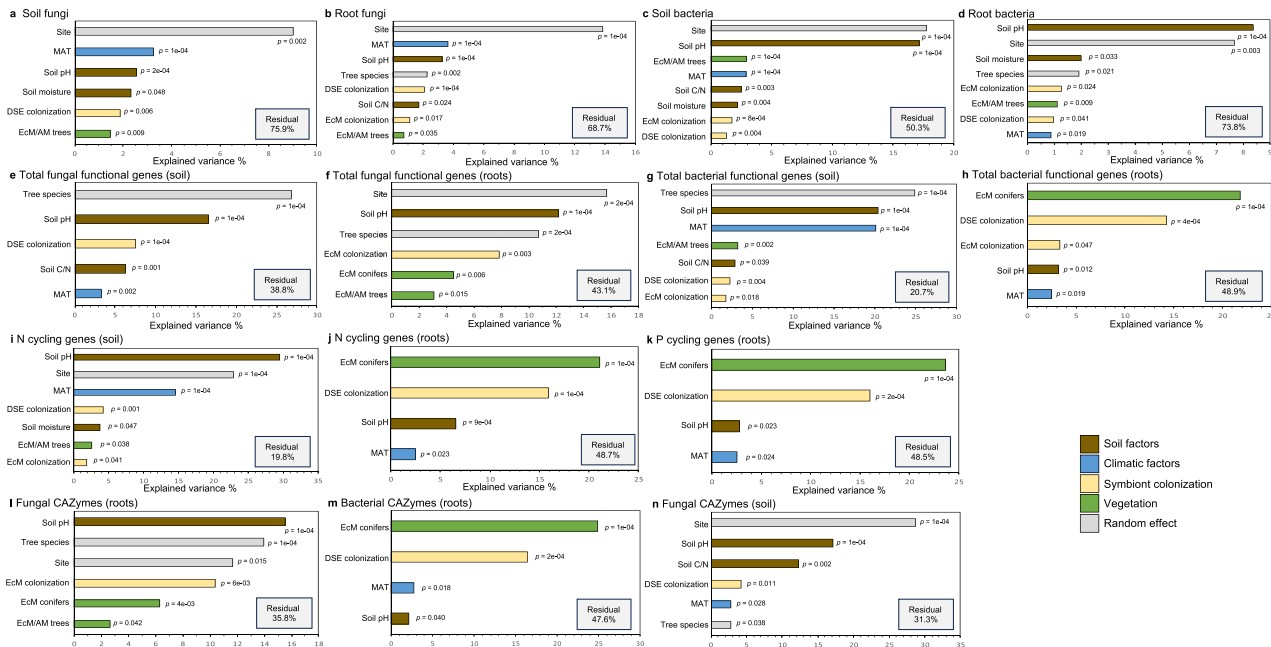

**Fig. 5 | Root symbioses play a potential role in structuring bacterial and fungal communities and their gene compositions in addition to climate, soil, and vegetation properties.** Results of permutational multivariate analysis of variance (PERMANOVA) (9999 permutations) on factors explaining the composition of (**a**) soil fungal communities ($n = 61$), (**b**) root fungal communities ($n = 61$), (**c**) soil bacterial communities ($n = 61$), (**d**) root bacterial communities ($n = 61$), (**e**) total fungal functional genes in soil ($n = 61$), (**f**) total fungal functional genes in roots ($n = 53$), (**g**) total bacterial functional genes in soil ($n = 61$), (**h**) total bacterial functional genes in roots ($n = 53$), (**i**) nitrogen (N) cycling genes in soil ($n = 61$), (**j**) N cycling genes in roots ($n = 53$), (**k**) phosphorus (P) cycling genes in roots ($n = 53$), (**l**) fungal carbohydrate-active enzymes (CAZymes) in roots ($n = 53$), (**m**) bacterial CAZymes in roots ($n = 53$), and (**n**) fungal CAZymes in soil ($n = 61$). Euclidean distances were used for centered-log ratio transformed bacterial and fungal operational taxonomic unit (OTU) tables from metabarcoding reads, and Bray-Curtis's

dissimilarity was used for normalized gene count tables from metagenomic reads. Soil factors (brown) are soil pH, soil carbon/nitrogen (C/N), and soil moisture; climatic factors (blue) are mean annual temperature (MAT), mean annual precipitation (MAP), and climatic moisture deficit (CMD); root symbiont colonization factors (yellow) are dark septate endophyte (DSE) colonization, ectomycorrhizal (EcM) colonization, and arbuscular mycorrhizal (AM) colonization; vegetation factors (green) are the relative basal area of EcM/AM trees, and the relative basal area of coniferous EcM trees. Variance explained by site and tree species as random effects is shaded gray, and residual variance is listed in the gray boxes. Only individual factors that were significant ($p < 0.05$) are displayed and their $p$-values are listed adjacent the factor ($p$ values were obtained using 9999 permutations), for detailed PERMANOVA results see Supplementary Data 4. The values on the x axis are different for each sub-figure. The statistical test used was two-sided.

Most notably for assimilatory (soil: $R^2$m = 0.27, $p$ = 8e-05, roots: $R^2$m = 0.20, $p$ = 2e-04) and dissimilatory nitrate reduction genes (soil: $R^2$m = 0.32, $p$ = 1e-04, roots: $R^2$m = 0.43, $p$ = 0.003), denitrification genes (soil: $R^2$m = 0.40, $p$ = 9e-06, roots: $R^2$m = 0.64, $p$ = 1e-09), nitrification genes (soil: $R^2$m = 0.22, $p$ = 0.002, roots: $R^2$m = 0.51, $p$ = 1e-04), and hydroxylamine reduction genes (soil: $R^2$m = 0.11, $p$ = 0.031, roots: $R^2$m = 0.40, $p$ = 1e-05).

Both DSE and EcM colonization showed significant relationships with the relative abundance of P cycling genes involved in specific processes in soil (Fig. S7; Supplementary Note 4), including negative relationships between EcM colonization and inorganic P mobilization genes ($R^2$m = 0.12, $p$ = 0.022) and DSE colonization with genes involved in P starvation response regulation ($R^2$m = 0.11, $p$ = 0.005). Only DSE colonization was consistently negatively associated with P cycling gene groups in roots (organic P mineralization: $R^2$m = 0.36, $p$ = 4e-05, inorganic P mobilization: $R^2$m = 0.21, $p$ = 0.001, and P starvation response regulation: $R^2$m = 0.31, $p$ = 3e-05) (Fig. S7).

### Both EcM and DSE colonization influenced tea bag decomposition
In terms of relationships between symbiont root colonization and rooibos tea bag decomposition, we found a positive correlation with DSE colonization ($R^2$m = 0.12, $p$ = 0.026) and a negative correlation with EcM colonization ($R^2$m = 0.21, $p$ = 0.010). Nevertheless, abiotic factors related to climate and soil and biotic factors such as specific CAZyme genes generally had stronger correlations with both rooibos

and green tea bag decomposition compared to root symbiont colonization (Fig. S8 and Supplementary Note 5).

### Potential DSE taxa, specific fungal CAZymes, and general functional genes associated with DSE colonization
As DSE colonization had a consistent association with various microbiome properties, our final step of analysis was to identify specific fungal taxa in roots from our list of potential DSE fungi (Supplementary Data 2) that were correlated with DSE colonization, as well as specific fungal CAZymes and general functional genes. The relative abundance of the genus *Cladophialophora* ($R^2$m = 0.28, $p$ = 2e-04) and specific OTUs from the genera *Cladophialophora* ($R^2$m = 0.41, $p$ = 2e-06) and *Phialocephala* ($R^2$m = 0.19, $p$ = 0.004) in roots were most strongly and positively correlated with DSE colonization (Fig. S9). For fungal CAZymes in soil and roots, DSE colonization was most strongly and positively correlated with the relative abundance of specific families of glycoside transferases (GT) in soil and glycoside hydrolases (GH) in roots (Fig. S9). In addition, for general functional genes (orthologous genes [OG]) in both soil and roots, DSE colonization was positively correlated with the relative abundance of various genes, including those encoding transposable elements such as retro-transposon proteins (Supplementary Data 5).

## Discussion
While climate and soil were generally the most important variables for explaining the properties of root and soil microbiomes, we show that DSE root colonization was consistently associated with the

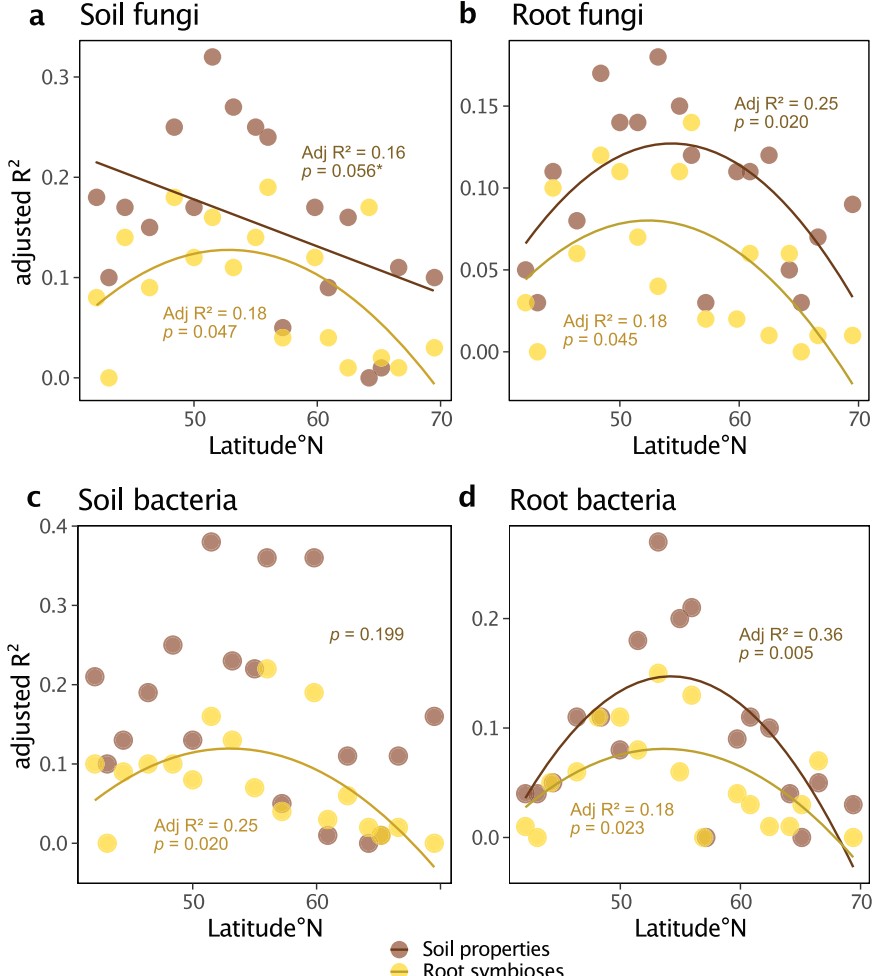

**Fig. 6 | The influence of root symbioses and soil properties on bacterial and fungal communities is related to latitude.** Linear and first order polynomial models of the total within site variance (adjusted $R^2$) in bacterial and fungal communities across latitude explained by root symbioses (arbuscular mycorrhizal, ectomycorrhizal, and dark septate endophyte colonization) and soil properties (pH, carbon/nitrogen, and moisture content) as calculated from variation partitioning analysis; (**a**) the composition of soil fungal communities (Euclidean distances on centered-log ratio transformed metabarcoding reads), (**b**) the composition of root fungal communities, (**c**) the composition of soil bacterial communities, and (**d**) the composition of root bacterial communities as explained by latitude. Euclidean distances were used for centered-log ratio transformed bacterial and fungal operational taxonomic unit (OTU) tables from metabarcoding reads. Soil properties are indicated by brown points and root symbioses are indicated by yellow points, for each model $n = 18$ (the number of independent sites), adjusted $R^2$ and accompanying $p$-values (calculated using the $t$-statistic from the $T$ distribution) are listed above for soil properties and below for root symbioses, only significant models ($p < 0.05$) or marginally significant models ($p = 0.05$–0.06) are displayed. The statistical test used was two-sided. For linear model summaries supporting this figure see Supplementary Data 12.

structure and potential function of root and soil microbiomes of widespread deciduous broadleaved tree species across a European latitudinal gradient. We also present evidence supporting the potential role of EcM colonization in structuring root and soil microbiomes across our studied sites and little evidence for the role of AM colonization. Our results further support previous findings that EcM associations are sensitive to moisture availability[30–32], and that AM associations are sensitive to soil pH and host tree basal area[13,33]. Despite EcM and AM associations being suggested to drive a wide range of ecological (soil) properties and processes [e.g[11,34–36,13]], we show that, at least under deciduous trees across a gradient of different climatic, soil, and vegetation properties, DSE colonization had stronger and more consistent associations with the soil and root microbiome and its functioning than EcM and AM colonization. Thus, root endophytic associations, particularly DSE, may be key biotic interactions explaining both soil and root microbial communities and their potential functions and consequently warrant further studies.

### The potential role of DSE in plant pathogen suppression

Increasing evidence suggests that DSE inhibit plant pathogens under controlled conditions[28,37–40] and our large-scale field study further supports this potential plant protective function. Specifically, DSE colonization was negatively associated with the relative abundance of putative fungal plant pathogens in soil and roots. The potential mechanisms underlying the suppression of plant pathogens by DSE may be by outcompeting pathogens for habitat space and plant-derived C and rhizosphere nutrients and by providing a barrier to pathogen colonization[25], a mechanism previously suggested for EcM[10]. Additionally, the potentially pronounced production of secondary metabolites by some DSE fungi compared to many other fungi[41] may directly antagonize pathogens by creating an inhibition zone or chemical barrier by direct antagonism to pathogen activity and by indirectly locking up nutrients necessary for pathogen growth, i.e., through the excretion of siderophores[28,37,42].

Several successful plant pathogens have expanded genomes with repeat-rich regions[43]. Interestingly, this is also found in several DSE

fungi, with highly repetitive transposable elements such as retrotransposons[44]. The similarity in genomic traits between plant pathogens and DSE fungi may explain their potentially latent parasitic function[45]. Yet, DSE are more often associated with healthy plants[27], hence their potentially increased abundance of transposable elements may explain their ecological flexibility and ability to outcompete pathogens when colonizing roots[9]. The protective function of DSE, and root endophytes in general, against plant pathogens in natural ecosystems warrants further study. This research could help fill gaps in plant-soil feedback research, which has focused primarily on mycorrhizal fungi.

## Root symbionts may influence root and soil microbiomes in addition to soil, climate, and vegetation

We found a negative association between DSE colonization and the bacteria/fungi ratio in soil and roots, although soil pH had a stronger association[46]. This ratio is a proxy for the dominance of bacteria and their key processes relative to fungi[47]. Support for DSE colonization in influencing bacterial communities was further indicated by negative relationships with bacterial diversity in soil and roots and total bacterial functional gene diversity in roots. Colonization by DSE also explained significant variance in bacterial communities and functional gene compositions in roots and soil alongside EcM colonization, EcM/AM trees, and EcM conifers; however, soil pH generally explained most variance, and MAT was also important. As we suggested for pathogens, DSE may impact bacteria by competing for habitat and nutrients and directly by excreting secondary metabolites[9]. Dark septate endophytes may also actively alter soil organic matter quality[48] and pH[49], which shape bacterial communities[50,51]. Our findings imply that soil pH and MAT best explain fungal communities and functional gene compositions. In addition, DSE and EcM colonization also accounted for variation in root and soil fungal communities and functional gene compositions. The influence of DSE fungi on fungal communities and their functioning may be related to their flexible symbiotic and free-living capabilities[26]. When other fungal guilds are limited by plant C or soil resources, DSE fungi may mediate interactions between other root and soil fungi, especially in harsh environments[37,52]. The effect of EcM colonization on fungal communities is more expected, and EcM fungi are generally the dominant fungal guild in EcM-dominated forests[13], such as the sites in this study. The interplay between DSE, EcM fungi, and other soil organisms appears to influence soil microbial community dynamics.

## Root symbioses and soil organic matter decomposition

Tea bag mass loss was best explained by climate (MAT and MAP) and the relative abundance of specific CAZyme groups. Furthermore, root symbioses directly affected tea bag mass loss and decomposer communities and their CAZymes. Climate is known as a major determinant of decomposition[53–55], which is not surprising given the temperature optimums of CAZymes[56,57], it also influences the biogeography of different types of root symbioses[1].

Growing evidence links EcM fungi to the dynamics of soil organic matter decomposition[58–61]. Our results show a negative correlation between EcM colonization and the mass loss of rooibos tea bags. Additionally, EcM colonization explained significant variance in the composition of fungal CAZymes in roots and negatively correlated with bacterial CAZymes in soil, especially those involved in cellulose degradation. The Gadgil effect[14,15] suggests that competition between EcM and saprotrophic fungi suppresses decomposition; but we did not find a negative relationship between EcM colonization and fungal saprotrophs, suggesting the interaction between EcM fungi and bacterial saprotrophs may be more important in this case.

Compared to EcM fungi, the role of DSE fungi in soil organic matter decomposition is less known. Dark septate endophytes may be relatively efficient degraders of organic matter due to their expanded repertoire of CAZymes, including plant cell wall degrading enzymes[62,63]. We found that DSE colonization positively associated with soil fungal CAZyme diversity and influenced their composition. We also found that DSE colonization was positively associated with the relative abundance of specific fungal CAZymes in soil and roots, such as glycoside transferases and glycoside hydrolases, which is consistent with previous findings that DSE genomes are enriched with such groups[62].

Additionally, DSE colonization was positively correlated with the relative abundance of soil saprotrophs and rooibos tea bag decomposition, supporting their potential influence on decomposition. Dark septate endophyte colonization also strongly correlated with the diversity of bacterial CAZymes, their composition in roots, and the relative abundance of particular bacterial CAZymes targeting various substrates. Through a rich arsenal of CAZymes targeting microbial necromass[64], which contributes considerably to soil organic matter[65], bacteria may complement fungi in degrading soil organic matter. Finally, DSE colonization was strongly linked with soil C/N, which DSE may reinforce through the slow turnover of their highly-melanized recalcitrant necromass[66]. Given our findings, the role of EcM and DSE fungi in soil organic matter decomposition warrants additional study to better understand their effect on soil C cycling.

## The potential influence of root symbioses on soil N and P dynamics

Both EcM and AM associations influence soil N cycling[10,11,21], but we found that DSE colonization strongly and consistently associated with the composition of N cycling genes and had a negative association with their diversity, compared to positive associations with EcM and AM colonization. Furthermore, DSE colonization was negatively correlated with the relative abundance of N cycling genes in soil and roots. Several studies suggest that DSE may help plants acquire N, especially under high C/N conditions where N becomes limited and DSE may access organically bound N[67,68]. Our results imply that DSE may impede soil N cycling by decoupling soil C and N, as suggested for root-associated ascomycetes in the Arctic[69]. This may reinforce high C/N conditions that locks plants into an N limitation loop, making DSE more valuable for plant N acquisition as suggested for EcM fungi[67].

Two of our tree species (*Alnus glutinosa* and *A. incana*) form obligate N-fixing actinorhizal associations with *Frankia* bacteria, which tend to undertake N fixation regardless of environmental conditions[70]. While we did not quantify *Frankia* root nodules, we found that the relative abundance of *Frankia* on roots was positively correlated with N cycling gene diversity and the relative abundance of N fixing genes. It is difficult to determine how much of the negative effect of DSE colonization on N cycling genes and bacterial-related microbiome properties is due to DSE presence or a lack of *Frankia*; however, our DSE results were generally robust across tree species, including non-N fixing *Sorbus* trees, and after accounting for the potential effect of *Frankia's* relative abundance on roots. The complex interactions between root symbionts require more research. As they form complex root associations with multiple symbionts, *Alnus* species can be used as a model to study how EcM fungi, AM fungi, root endophytes and N fixing bacteria interact on the same root systems.

We also found that DSE colonization explained a large proportion of variance in P cycling gene compositions and negatively correlated with the relative abundance of specific P cycling genes. By reducing pH, DSE fungi may help plants acquire sparingly soluble P in the rhizosphere[49]. Our findings suggest that roots with higher DSE colonization may have a reduced P cycling capacity, as shown by the lower relative abundance of P cycling genes. Our results more strongly link DSE colonization with soil N and P cycling genes compared to EcM and AM colonization, which is surprising considering the well-established role of mycorrhizal associations in plant-soil nutrient cycling.

## The potential effect of soil properties and root symbioses on microbial communities varies with latitude

While we have focused mainly on the potential effect of root symbioses on soil and root microbiome properties, our results indicate that climate and soil properties are important for explaining the colonization and relative abundance of different root symbionts, the structure of bacterial and fungal communities, and their functional genes. Climate is known to shape the distribution of different root symbioses[1], and together with soil affects the structure and function of bacterial and fungal communities[71–74]. Using latitude as a proxy for climate and environmental gradients, we showed that root symbioses (AM, EcM, and DSE colonization) and soil properties (pH, C/N, and moisture) explained more variance in root and soil microbial communities at mid-latitudes. This suggests a climatic control of plant-soil-microbial interactions[75], particularly in more extreme environments, and has implications for predicting the response of plant-soil systems to environmental change factors.

## The ambiguity of fungi forming DSE

We have presented extensive evidence that DSE potentially shape the structure and function of root and soil microbiomes via root colonization rates. However, DSE as a fungal guild of investigation is ambiguous. The taxonomy and phylogenetic breadth of fungi forming DSE associations is poorly understood[76], there is no highly curated database specifically for DSE taxa, and DSE are usually assigned at the strain level. Furthermore, the ecology and functioning of DSE is obscure[25,26], with cryptic lifestyles often closely associated with EcM, AM, and ericoid mycorrhizal associations on woody plants, and they may form different associations on different hosts, reflecting structural and functional flexibility[77–80]. With caution, we found that DSE colonization via microscopy correlated with the relative abundance of putative DSE taxa in roots from genera-based metabarcoding annotations. Two fungal OTUs, a *Cladophialophora* species and a *Phialocephala* species, had the strongest correlations, suggesting they are ubiquitous DSE generalists colonizing tree roots. Strains of *Cladophialophora* and *Phialocephala* species are associated with mitigating disease, altering plant nutrient uptake, and plant growth promotion[28,40,81–83]. To better understand DSE associations, their importance, and potential applications, we need more research characterizing, isolating, culturing, and re-inoculating plant roots with DSE. Long-read sequencing of field-colonized plant roots may help uncover potentially relevant DSE taxa at the strain level.

This study employed metabarcoding, metagenomics, and more direct measurements (e.g., microscopy and tea bag mass loss) to show how root symbionts, particularly DSE, may shape soil and root communities and processes. Although functional guild annotation based on metabarcoding data has limitations, root colonization by different symbionts (obtained by microscopy) and the relative abundance of sequences obtained by metabarcoding was correlated in our study. The relative quantification of root symbionts by metabarcoding and microscopy also showed similar associations with environmental variables.

## Implications

Our findings on the potentially important role of DSE associations in structuring soil and root microbiomes and their functions have various insights for future research. It highlights the need for further studies on the ecological functions of DSE and how tree-root symbioses affect soil communities and processes in relation to different host-plant characteristics, climate, and environmental properties. The consistent association between DSE with root and soil fungal and bacterial communities and processes of broadleaved tree species from three genera suggests the need to assess their importance in more tree species with different traits, such as conifers, and to quantify microbial community functioning using metatranscriptomic and metabolomic approaches

linked to metabarcoding and metagenomics. Since our study is limited to boreal and temperate ecosystems, it remains to be shown if DSE, and root endophytic fungi in general, have similar associations with plant-associated microbiome properties in tropical ecosystems, which have vastly different climates and nutrient cycling regimes.

A warmer and more fluctuating climate may favor AM and N-fixing trees, their associated soil communities, and plant-soil feedbacks[84,85]. Thus, under future climate regimes, AM fungi, pathogens and saprotrophs may increase in prevalence[75], EcM fungal communities may change including a reduction in their biomass[3,86], which may affect forest productivity[87]. Our findings imply that root endophytic fungi such as DSE may also play important roles in these systems, warranting additional exploration of how climatic changes alter their abundance and interactions with plants and soil. Our study emphasizes the importance of understanding the complex interactions of root symbionts with trees and soil microbial communities for attenuating the impact of environmental perturbations on forest ecosystem functioning.

## Methods

### Sampling sites and host tree selection

The study was conducted across a 3220 km latitudinal gradient, which included 18 locations that were either established research stations or vegetation reserves from northern Norway (69.46°N, 30.02°E) to central Italy (42.08°N, 9.82°E) (Fig. 1a). Mean annual temperature (MAT) ranged between 0.8 and 15.1 °C, mean annual precipitation (MAP) between 486 and 1213 mm, and altitude between 7 and 1211 m above sea level (Supplementary Data 1). Selected sites contained widespread deciduous broadleaf tree species from three genera (*Alnus*, *Betula*, and *Sorbus*), differing in their preference for different mycorrhizal associations and N-fixing status (*Alnus*: EcM/AM/N-fixing, *Betula*: EcM, and *Sorbus:*AM). *Alnus incana* was present at 13 sites from northern Norway to northern Italy; *A. glutinosa* at 12 sites from central Sweden to central Italy; *Betula pendula* was present at all sites; and *Sorbus aucuparia* was present at all but the two southernmost sites in Italy, where it was replaced by *S. torminalis* or *S. domestica* (Fig. 1a). At each site, 5 trees of each target species were selected with a diameter at breast height (DBH) of 10 and 20 cm, and a distance of at least 10 m within a 50 m radius of one another. If different target species were growing separately (> 50 m) or in clearly different habitats, we considered these habitats as separate plots, resulting in 11 sites with 1 plot, 3 sites with 2 plots, 3 sites with 3 plots, and 1 site with 4 plots, leading to a total of 30 plots across the 18 locations. In each plot (2500 m²), all tree species (target and non-target) > 10 cm DBH were recorded; their relative basal areas were estimated based on DBH measurements and assigned a mycorrhizal type according to the FungalRoot database[88]. Sites generally were EcM-dominated mixed forests (60 and 95% relative basal area).

### Sampling method

At each site, soil and root samples were collected from each of the selected trees (5 trees per tree species per site) by the same person with help from others between the 5th of August (Norway) and the 11th of September 2019 (central Italy) from sites in a north-south direction to roughly approximate a similar later stage in the growing season. Around each tree, four soil cores (5 cm in diameter and 10 cm in depth, with no distinction made between mineral and organic layers) were taken from each cardinal direction 50 cm from the base of the tree after removing loose litter and pooled into one composite sample per tree. Subsamples of each soil sample were air dried at room temperature for molecular analysis or further dried at 70 °C for 48 h for physicochemical analysis. Fine roots (<2 mm) were collected from the top 10 cm of soil by tracing them to the major lateral roots of each target tree, which were cleaned with water and stored in 70% ethanol until further analysis. Additionally, at 10 sites from Norway to

 

Germany, one Lipton teabag with rooibos (C/N = 43) and one with green tea (C/N = 12) were buried to 10 cm deep at one soil sampling point per tree and collected after 12 months according to a modified version of the Teabag Index protocol[89]; a total of 284 out of 330 teabags were recovered (leaving at least three replicates per species per site) and dried at 70 °C until constant weight to measure mass loss. All samples were collected from locations in the public domain or in previous agreement with the local community and/or property owner.

## Root colonization analysis

The root colonization rate (% of root tips colonized) of EcM fungi was assessed on *Betula* and *Alnus* fine roots under a dissecting microscope at ×20 magnification. Cleaned roots were cut into 1 cm pieces (total around 20 cm), spread randomly in a petri dish with water, and a total of 100 tips were scored EcM if a mantle was present (swollen short-root tips covered in hyphae with root hairs absent) or uncolonized if not present. Each sample of roots was analyzed twice after the random rearrangement of root sections. The root colonization rate (% of root length colonized) of AM fungi and DSE was assessed on *Alnus* and *Sorbus* fine roots by the grid line intersection method[90]. First, 1 cm pieces of cleaned roots (around 10 cm total) from each tree sample were cleared in a 2.5% KOH solution and stained with trypan blue in an acidic glycerol solution[91]. Stained roots were then mounted on slides, and colonization was estimated under a minimum of 100 random fields of intersection by using a ZEISS Axioscope 5 Digital Microscope (Carl Zeiss Microscopy, Oberkochen, Germany) at ×400 magnification. For AM fungi, the presence of stained arbuscles, hyphae, and vesicles (if present) were counted separately, and DSE structures were counted when darkly pigmented septate hyphae and clusters of inflated, rounded, and thick-walled cells (microsclerotia) were present within the cortical cells of roots[92] (for photographic examples of DSE structures in our root samples, see Fig. S10).

## Soil chemical analysis

Gravimetric soil moisture was obtained from 25 g of field moist soil per sample, followed by oven drying (70 °C) until a constant weight was reached and re-weighed to calculate the moisture content (%) of field moist soil. Soil pH, available phosphorus (P), potassium (K), calcium (Ca), magnesium (Mg), aluminum (Al), and iron (Fe) were analyzed after extraction in ammonium lactate (AL) solution at Agrilab (Uppsala, Sweden). Soil $^{13}C$ and $^{15}N$ natural abundances and total soil C and N contents were determined with an Isotope Ratio Mass Spectrometer/ IRMS (delta V Advantage, Thermo-Fisher Scientific, Dreieich, Germany) coupled to an Elemental Analyzer (Euro EA, Eurovector, Milano, Italy).

## Climate data

Climate data was obtained for geographic coordinates from CHELSA V2.1[93]. We also obtained the annual climatic moisture deficit (CMD) from INDECIS (https://indecis.csic.es/) as a proxy of water availability, which is the difference between atmospheric evaporative demand and precipitation[94].

## Metabarcoding of DNA from soil and root samples

Total soil DNA was extracted from 250 mg of air-dried and pulverized soil from each sample using the DNeasy PowerSoil Pro Kit (Qiagen GmbH, Hilden, Germany) following the manufacturer's instructions. Total root DNA was extracted from 15 mg of air-dried and pulverized roots from each sample using the Mag-Bind® Plant DNA DS 96 kit (Omega Bio-Tek, Norcross, GA, USA) according to the manufacturers' instructions and a Kingfisher™ Flex extraction robot (Thermo-Fisher Scientific), carried out by the Centre for Genetic Identification (CGI), Swedish Museum of Natural History, Stockholm, Sweden. The quality of DNA was checked based on 260/280 and 260/230 nm wavelength ratios using a NanoDrop 1000 spectrophotometer (Thermo-Fisher

Scientific). The quantity of DNA was measured using a Qubit dsDNA BR Assay Kit and the Qubit 2.0 Fluorometer (Thermo-Fisher Scientific).

Root DNA samples were subjected to metabarcoding by the SLU Metabarcoding Laboratory (UMBLA, SLU, Uppsala, Sweden) and soil DNA metabarcoding was performed by the authors. Metabarcoding of bacteria in both roots and soil was performed using the primers 515FB and 926R to target the ribosomal rRNA 16S gene V4-V5 regions[95]. The ITS2 subregion was targeted for soil fungi using the primers gITS7ngs and ITS4ngsUni[96], and for root fungi using the primers gITS7[97] and ITS4[98]/ITS4arc (adapted for Archaeorhizomycetes[59]).

For soil ITS, soil 16S, and root ITS samples, all primers had unique 8 or 10 bp identification tags attached for bioinformatic identification of individual samples after pooling. Both ITS2 and soil 16S samples were amplified in duplicate 50 µl PCR reactions consisting of: 8.25 µl of ddH$_2$O, 5 µl of 10X DreamTaq Buffer (Thermo-Fisher Scientific), 5 µl of dNTPs (2 mM), 1.5 µl of MgCl$_2$ (25 mM), 0.25 µl DreamTaq Green DNA Polymerase (5 units/µl) (Thermo-Fisher Scientific), 2.5 µl of both forward and reverse primers (0.4 µM; for roots 3/4 ITS4 at 0.3 µM and 1/4 ITS4arc at 0.1 µM), and 25 µl of template DNA (1 ng/µl) according to[99]. Thermal cycling conditions were as follows: 95 °C for 15 min, 30 cycles of 95 °C for 30 s, 56 °C for 45 s, and 72 °C for 1 min, with a final extension at 72 °C for 10 min.

Root 16s samples were amplified using a two-step PCR procedure in duplicates; the first consisted of 10 ng extracted DNA, 1× Phusion PCR Mastermix (Thermo-Fisher Scientific), 1 mg/ml BSA, and 0.25 µM of the primers 515FB and 926R to target the ribosomal rRNA 16S gene V4-V5 regions[95] in 15 µl reactions amplified under the following conditions: 3 min at 98 °C, followed by 25 cycles of 98 °C for 15 s, 50 °C for 30 s, and 72 °C for 40 s, and a final extension step of 10 min at 72 °C. The PCR products were then pooled and checked by agarose gel electrophoresis, followed by cleaning with Sera-Mag bead purification (Cytiva Life Sciences, Marlborough, MA, USA). A single 30 µl reaction was performed for the second PCR, using 0.2 µM of primers with Illumina Nextera adaptor and index sequences and 3 µl of the pooled PCR product from the first PCR. Conditions were the same as the first step, except for an annealing temperature of 55 °C and an extension time of 45 s with 8 cycles.

All PCR products were size checked by running 2 µl of DNA on a 1% agarose gel for 15 to 20 min. The duplicate PCR amplicons were then pooled and cleaned using Sera-Mag bead purification (Cytiva Life Sciences) according to the manufacturer's instructions, then all samples were pooled into an equimolar mix of 100 to 105 samples and cleaned with the E.Z.N.A. Cycle Pure Kit (Omega Bio-Tek), followed by a quality check on an Agilent Bioanalyzer (Agilent, Santa Clara, CA). Pools were finally shipped for library preparation and 250 PE sequencing on the Illumina NovaSeq 6000 Sequencing System (Illumina, San Diego, CA, USA) at the service provider's (Novogene Europe) laboratory.

## Bioinformatics of amplicon sequences

The LotuS2 pipeline[100] was used to quality filter, demultiplex, and process filtered reads into operational taxonomic units (OTUs). The ITS2 region was extracted, and non-ITS sequences were eliminated using ITSx[101]. Uchime was used for chimera detection and removal[102], and all singletons and sequences shorter than 100 bp were discarded. Clustering of sequences was done using a de-novo clustering algorithm in UPARSE[103] with a 97% similarity threshold, and taxonomy was assigned against the SILVA and UNITE databases for prokaryotic and fungal sequences, respectively. All OTUs representing archaea, chloroplasts, eukaryotes, and mitochondria were omitted from the bacterial dataset, and OTUs not assigned to fungi were omitted from the ITS dataset after OTU clustering. This resulted in a total of 5 326 931 and 3 272 525 reads, respectively covering 9772 fungal and 8142 bacterial OTUs for soil samples, and 10 511 905 and 23 407 319 reads, respectively covering 4322 fungal and 19 355 bacterial OTUs for root samples.

Two soil 16S and ITS samples, four root 16S samples, and 24 root ITS samples yielded insufficient reads or failed to amplify and were excluded from further analysis. Fungal OTUs were subsequently assigned to a primary lifestyle at the genus level using the FungalTraits tool[29]. In addition to the primary lifestyle alignments, we assigned a secondary lifestyle to potential DSE fungi (genus level) based on genera that contain species or strains with potential DSE capacity based on the category 'root endophytic capacity' in the FungalTraits tool (for a list of the potential DSE taxa used in this study see Supplementary Data 2).

## Shotgun metagenomics and bioinformatics

Shotgun metagenomic sequencing was performed on pooled equimolar amounts of DNA from five replicate soil and root samples per tree species per site. All 61 composite soil DNA pools (*Alnus glutinosa* $n = 12$, *A. incana* $n = 13$, *Betula pendula* $n = 18$, *Sorbus aucuparia* $n = 16$, *S. domestica* $n = 1$, and *S. torminalis* $n = 1$) passed initial quality control checks and proceeded to library preparation. Eight out of 61 composite root DNA pools failed quality control checks and were excluded, leaving 53 pools for library preparation (*Alnus glutinosa* $n = 11$, *A. incana* $n = 11$, *Betula pendula* $n = 14$, *Sorbus aucuparia* $n = 15$, *S. domestica* $n = 1$, *S. torminalis* $n = 1$). Library preparation and sequencing were performed at the service provider facilities (Novogene Europe) using the Novogene NGS DNA Library Prep Set kit and sequenced on Illumina NovaSeq with $2 \times 150$ bp paired end reads, with an average of 34 625 862 ($\pm$ 6 134 210) reads per sample for soil and 33 456 982 reads for roots ($\pm$ 5 292 126) (Supplementary Data 6).

Analysis of metagenomic reads was done using the MATAFILER pipeline[104], following a bioinformatic strategy developed in[71]. Briefly, reads obtained from the shotgun metagenomic sequencing of soil and root samples were quality-filtered by removing reads shorter than 70% of the maximum expected read length (150 bp), with an observed accumulated error >2 or an estimated accumulated error >2.5 with a probability of ≥0.01, or >1 ambiguous position. Using sdm software (version 1.46)[105], reads were trimmed if base quality dropped below 20 in a window of 15 bases at the 3′ end or if the accumulated error exceeded 2. Altogether, 61 soil samples (between 27 950 077 and 56 986 895 total reads per sample) and 53 root samples (between 22 600 117 and 51 501 361 total reads per sample) produced a sufficient quantity of reads and were retained for statistical analyses. To estimate the functional composition of each sample, we implemented a similarity search approach using DIAMOND (version 2.0.5; options -k 5 -e 1e-4−sensitive) in blastx mode[106]. Prior to that, the quality-filtered read pairs were merged using sdm. The mapping scores of two unmerged query reads that mapped to the same target were combined to avoid double counting. In these cases, the hit scores were combined by averaging the percent identity of both hits. The best hit for a given query was based on the highest bit score and highest percent identity to the subject sequence. Using this method, we calculated the relative abundance of (clusters of) orthologous gene groups (OG) by mapping quality-filtered reads against the eggnog database (version 4)[107], reads were also mapped against the KEGG (Kyoto Encyclopedia of Genes and Genomes) database[108], and for functional specific carbohydrate-active enzyme (CAZyme) annotations, we mapped reads against the latest version of the CAZy database[109]. We further grouped selected CAZymes (Supplementary Data 7) based on their active substrates, including plant biomass (cellulose, hemicellulose, and lignin), fungal biomass (chitin and glucans), and bacterial biomass (peptidoglycan), according to the literature[110,111] for analysis of relative abundances. For all databases that included taxonomic information (eggNOG, KEGG, and CAZyme), reads were mapped competitively against all kingdoms and assigned into prokaryotic and eukaryotic groups on the basis of the best bit score in the alignment and the taxonomic annotation provided within the database at the kingdom level.

For diversity of N cycling genes, we focused on those (sub)families listed in the NCycDB database[112] (Supplementary Data 8). For relative abundances, we grouped genes into the following N cycling gene pathways, also according to the NCycDB: nitrogen fixation, nitrification, denitrification, assimilatory nitrate reduction, dissimilatory nitrate reduction, organic N degradation, organic N synthesis, and hydroxylamine reduction. For diversity of P cycling genes, we focused on those we could match with KEGG IDs in the PCycDB database[113] (Supplementary Data 9). For relative abundances, we focused on genes involved in the following P cycling pathways: organic P mineralization, inorganic P solubilization, and P-starvation response regulation, grouped according to the literature[114].

We also calculated metagenomic relative abundances (i.e., miTag[115]) of different taxonomic groups, including the metagenomic bacteria/fungi ratio, based on small subunit (SSU) rRNA genes. For this, SortMeRNA (version 2.0)[115] was used to extract and blast search rRNA genes against the SILVA SSU database (v128). Reads approximately matching this database with $e < 10^{-4}$ were further filtered with custom Perl and C++ scripts[116] and merged using sdm. In cases where read pairs could not be merged, the reads were interleaved such that the second read pair was reverse complemented and then sequentially added to the first read. Of these preselected reads, 50,000 reads were fine-matched in the Silva SSU database using Lambda and the lowest common ancestor (LCA) algorithm adapted from LotuS[100].

## Data analysis

Data management, statistical analyses and data visualizations were primarily done using the R platform[117] v.4.0.3 and primarily the following packages: *phyloseq*[118] v.1.22.3, *microbiome*[119] v.1.20.0, *vegan*[120] v.2.6.2[121], v.1.0.9, *ggplot2*[122] v.3.3.6, *Hmisc*[123] v.4.7.0, *corrplot*[124] v.0.92, *adespatial*[125] v.0.3.16, *randomForest*[126] v.4.7.1, *piecewiseSEM*[127] v.2.1.2, *lme4*[128] v.1.1.32, *lmerTest*[129] v.3.1.3, *ggeffects*[130] v.1.2.1, and *MuMIn*[131] v.1.46.0. Additional analysis was performed in the Primer v7 software (PRIMER-e Quest Research Limited, Auckland, New Zealand) using the PERMANOVA+ routine[132].

The Shannon diversity (H) index was calculated using the *vegan* package on rarefied OTU and gene counts to account for differences in sequencing depth. Both ITS and 16S samples were rarefied to 2000 reads for soil and 5000 reads for roots. For the functional gene categories, total functional genes (OG), CAZymes, P cycling, and N cycling gene counts were rarefied to a sampling depth of 27 950 077 reads for soil or 22 600 117 reads for roots. For the analysis of the relative abundance of different taxa or functional guilds, raw OTU counts were centered-log ratio (clr) transformed with an added pseudo count using the *microbiome* and *phyloseq* packages. By comparing read abundances to the geometric mean, this method is less sensitive to sequencing depth and accounts for the compositional nature of microbial high-throughput sequencing data while retaining relative and absolute abundance information[133]. The relative abundance of potential DSE taxa was calculated relative to all other fungi separately to the primary fungal lifestyle assignments, meaning that these taxa were included in other lifestyles within the relative abundance calculations for primary lifestyles. For the analysis of functional gene relative abundances from shotgun metagenomic data, all abundance matrices were normalized as a percentage of the total number of reads used for mapping; this takes into account differences in library size and has the advantage of including the fraction of unmapped (functionally unclassified) reads[71]. All statistical analyses were performed on either the mean from at least 3 replicate tree individuals (metabarcoding data) or composite (shotgun metagenomic data) value per tree species per site, unless otherwise specified. We also calculated and reported the standard error for all mean values, where applicable.

To account for multicollinearity of biotic and abiotic predictor variables, we removed variables with a Spearman's rank correlation > 0.7[134], while also considering the most important variables for answering our hypotheses. In our final analysis, we included the climatic variables of MAT, MAP, and CMD, the soil variables of pH, C/N, gravimetric soil moisture content, $\delta^{15}N$, $\delta^{13}C$, and P, and the vegetation variables of tree diversity (Shannon diversity index), EcM/AM tree basal area, and coniferous EcM tree basal area. We were interested in EcM, AM, and DSE root colonization as both response and predictor variables.

For soil and root microbiome response variables, we focused on the diversity of bacteria and fungi (metabarcoding), the ratio of bacteria/fungi (metagenomics), the diversity of total bacterial and fungal functional genes, CAZymes, P cycling genes, and N cycling genes (metagenomics). For the relative abundance of different fungal guilds, we focused on soil saprotrophs, plant pathogens, and the symbiotic guilds of EcM, AM, and potential DSE fungi (for a list of potential DSE taxa, see Supplementary Data 2). We also considered the relative abundance of the bacterial genus *Frankia* (metabarcoding) and the relative abundance of specific bacterial and fungal CAZyme, P cycling, and N cycling gene groups based on process or substrate (metagenomics). And finally, we analyzed the mass loss of rooibos and green tea bags.

To get a general idea of the strength and direction of relationships between the abiotic and biotic predictors of root colonization, tea bag decomposition, and our soil and root microbiome properties of interest, we used a combination of the random forest algorithm, a machine learning approach[126], using the *randomForest* package, together with Spearman rank correlation analysis using the *corrplot* package, and we adjusted *p* values for multiple comparisons using the Benjamini & Hochberg method[135], followed by forward selection using the *adespatial* package. To test individual relationships and to account for the effect of site, plot, and tree species, we fitted linear mixed-effects models using the *lme4* package with plot embedded in site crossed with tree species as a random effects structure, unless a random effect explained zero variance, in which case they were removed from a given model. Marginal (fixed effects only) and conditional (random and fixed effects) $R^2$ values for each model were calculated using the *MuMIn* package, and *p* values were calculated in the *lmerTest* package using the Satterthwaite approximation. To assess model fit, compare models, and select the best fitting models, we evaluated standardized model residuals, calculated and compared the Akaike information criterion (AIC), and considered *p* values and $R^2$ values. Fitted models and their standard error were plotted using the *ggeffects* package.

To examine the robustness of the significant relationships observed between root colonization by different root symbionts and microbiome properties while accounting for the effect of abiotic and other biotic factors together with direct and indirect effects, we performed structural equation modeling (SEM) using the *piecewiseSEM* package incorporating linear mixed-effects models using the *lme4* package with plot embedded in site crossed with tree species as a random effects structure, and the Kenward-Roger approximation was used to calculate the significance of individual paths. We determined the best fitting models based on AIC values using a backwards selection approach and achieved optimal model fit through subsequent iterative revisions based on modification indices.

For the multivariate analysis of factors explaining variation in bacterial and fungal communities and functional gene compositions (composite values per tree species per site), we performed permutational multivariate analysis of variance (PERMANOVA) using PERMANOVA+ in Primer v7. For this analysis, we created distance matrices for bacterial and fungal communities using Euclidean distances on clr transformed data and the Bray-Curtis dissimilarity on normalized gene count matrices. We specified site and tree species as random effects and the climatic factors MAT, MAP, and CMD, the soil factors pH, C/N,

soil moisture, and the root symbiotic factors DSE, EcM, and AM colonization as fixed effects. The analysis was performed using 9999 permutations with a sequential sum of squares for the construction of the pseudo-*F* test statistic and its statistical significance. The same variable order was used for each response matrix (Supplementary Data 4). In the case that an explanatory variable had a negative estimated component of variation, we removed them from the model and reperformed the analysis, and the proportion of variance explained by each component (both fixed and random effects) was calculated by dividing each estimated component of variation by the total variance explained in the model, including residual variance.

For the multivariate analysis of factors explaining variation in bacterial and fungal community compositions (roots and soil) within sites (using all site-level tree sample replicates for each tree species), and to explore the effect of latitude on the extent of variation explained by soil properties and root symbioses, we performed variation partitioning analysis using the *vegan* package. For this analysis, we partitioned variance in bacterial and fungal community distance matrices (Euclidean distances from clr transformed abundance matrices) between soil properties (composite of soil pH, C/N, and moisture), root symbioses (composite of AM, EcM, and DSE colonization), and tree species at each site. We then plotted the adjusted $R^2$ for the total explained variance by the components of soil properties and root symbioses against latitude. Then for each component, we fitted simple linear or first-order polynomial models in the case of unimodal relationships (selected based on AIC values).

### Reporting summary

Further information on research design is available in the Nature Portfolio Reporting Summary linked to this article.

### Data availability

The 16S and ITS metabarcoding data generated in this study have been deposited in the NCBI Sequence Read Archive (SRA) under the Bio Project accession number: PRJNA1021963. All root and soil metagenomic sequences and associated metadata have been deposited in the European Nucleotide Archive (ENA) under accession number: PRJEB56450. The carbohydrate active enzymes (CAZy) Database is available under http://www.cazy.org/, the nitrogen cycling gene (NCyc) Database is available under https://github.com/qichao1984/NCyc [https://doi.org/10.1093/bioinformatics/bty741], the phosphorus cycling gene (PCyCDB) Database is available under https://github.com/ZengJiaxiong/Phosphorus-cycling-database [https://doi.org/10.1186/s40168-022-01292-1]. The Evolutionary genealogy of genes: Non-supervised Orthologous Groups (EggNOG) Database is available under http://eggnog5.embl.de/#/app/home. The Kyoto Encyclopedia of Genes and Genomes (KEGG) Database is available under https://www.genome.jp/kegg/kegg1.html. The data supporting the results and figures in this study are provided in the Supplementary Information/Source Data file. The FungalTraits tool used for fungal guild assignments is available under https://docs.google.com/spreadsheets/d/1cxImJWMYVTr6uIQXcTLwK1YNNzQvKJJifzzNpKCM6O0/edit?usp=sharing [https://doi.org/10.1007/s13225-020-00466-2]. The extended data including raw metabarcoding OTU tables, functional gene matrices (metagenomics), root symbiont colonization rates, tea bag mass loss, and environmental metadata generated in this study are deposited in the Zenodo open data repository (CERN) under https://doi.org/10.5281/zenodo.10203817. Source data are provided with this paper.

### Code availability

The pipeline to process metabarcoding samples is available under https://lotus2.earlham.ac.uk/main.php?site=downloads [https://github.com/hildebra/lotus2].The pipeline to process shotgun

metagenomic samples is available under https://github.com/hildebra/MATAFILER [https://doi.org/10.5281/zenodo.5831723].

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

## Acknowledgements

This study was primarily funded by the Swedish Research Council (Vetenskapsrådet; grants: 2017-05019 & 2021–03724 awarded to M.B.). We thank Katarina Ihrmark for her guidance in the molecular lab, and Benjamin Herold and Helena Klöckener for their assistance with field-work. We thank Paul Eric Aspholm and NIBIO Svanhovd, Charlotta Erefur and the SLU Svartberget SITES Research Station, Krister Karlsson and the SLU Siljanfors Research Station, Mikael Andersson, and the SLU Asa SITES Research Station, Kristian Graubæk and The Danish Nature Agency, Benjamin Herold and the Schorfheide-Chorin Biosphere Reserve, Axel Pampe and the Lower Saxony Forestry Office Reinhausen, and Carsten Mueller and the TUM School of Life Sciences Freising for facilitating the location of sampling sites, access and sampling permissions. J.F. and F.H. were supported by the Biotechnology and Biological Sciences Research Council (BBSRC) Institute Strategic Program Gut Microbes and Health BB/r012490/1 and its constituent project BBS/e/F/000Pr10355, Core Capability Grant BB/CCG1720/1 and the work delivered via the Scientific Computing group, as well as support for the physical HPC infrastructure and data centre delivered via the NBI Computing infrastructure for Science (CiS) group. F.H. was also supported by European Research Council H2020 StG (erc-stg-948219, EPYC). J.O. was supported by the Estonian Research Council (PSG784).

## Author contributions

T.N. designed the study together with M.B., J.B., and E.J.K.; T.N. performed the field data collection, soil molecular analysis, and EcM colonization analysis. J.O. performed the AM and DSE colonization analyses. F.B. and K.P. provided expertize and analysis of stable isotopes and total C and N measurements. M.B. performed metabarcoding bioinformatics. F.H. and J.F. performed shotgun metagenomics bioinformatics, T.N. performed the data analyses with contributions from M.B.; T.N. led the writing of the manuscript with contributions from all authors.

## Funding

## Competing interests

The authors declare no competing interests.
