## [Peer Review File · Nature Communications]

Pervasive associations of dark septate endophytic fungi with tree root and soil microbiomes across EuropeReviewers' comments:

Reviewer #1 (Remarks to the Author):

Overall, this paper presents the results a large scale field sampling campaign. They leverage metabarcoding studies of bacteria and fungi to understand correlations between different mycorrhizal types, the soil microbiome and various climatological variables. This study is well conducted, but fails to meet the mark of novelty and significant insight that I have come to expect from Nature Communications.

Specific elements where this manuscript can be improved include delineation of the hypotheses. My understanding is that there is already decent evidence for Hypothesis 1, from global topsoil surveys. In particular hypotheses 2 and 3 are not aligned with the evidence provided by the introduction. For example, what does temperate mean in this case? It is poorly operationalized in this hypothesis and throughout the paper.

Positive elements of this paper include the work done to study the percent colonization of various symbionts and the overall methods, including the bioinformatics are state of the art. Besides the teabags decay experiment, I am missing a functional characterization of the soil communities that would aid in the interpretation and significance of the work.

Finally, the title of the paper is misleading, as it is unclear why warming would lead to any changes based on the evidence at hand. The title needs to be rewritten to encapsulate the main findings of the study, which are limited in their capacity to predict future microbiome activity and function in forest landscapes

Reviewer #2 (Remarks to the Author):

Netherway and colleagues examined how mycorrhizal associations affect surrounding soil microbiomes and ecosystem function across a latitudinal gradient in Europe. They found strong effects of EM and DSE fungi but smaller effects of AM fungi on the relative abundance of soil saprotrophs, pathogens and decomposition. In general, effect sizes were larger in temperate compared to boreal forests. This work is a nice overview of fungal effects belowground across climatic gradients and one of the first to include DSEs as well as AM and EM mycorrhizal fungi. However, I have several comments on the novelty, hypotheses, analysis, and conclusions of this work. Detailed comments are below.

Novelty: Several studies have targeted similar questions. Here it would be helpful to distinguish how the current work is different from previous studies on microbiomes associated with different mycorrhizal types (e.g., Bahram et al. 2018) or those focused on biogeochemical differences among mycorrhizal types.

Hypotheses: Overall the hypotheses are difficult to follow in the results and discussion sections. It would be helpful to include language in both sections about testing certain hypotheses and if they were rejected.

In addition, the hypotheses could be clearer by pointing out the contrasting conditions you've studied. For example, why would temperate forests have the highest impact? Compared to what other ecosystem(s)?

Is colonization reflective of the strength of symbiosis? What if the colonizing fungi are more parasitic than mutualistic?

Statistics. It is unclear how the linear models accounted for the hierarchical design of the plots and

sites. Each model should have plot within site as a random effect. Do the Spearman rank correlations include this information?

Colonization v. molecular relative abundance: It's unclear why both methods were performed. In addition, how was the relative abundance of the different fungal groups calculated in the soil? Are the databases good enough to define DSEs accurately? How many taxa were assigned to each group? If most of the sequences were assigned to a fungal functional group, then relationships in each focal group could be caused by the change in another group, not necessarily the focal group.

References

Bahram et al. 2018. Structure and function of the global topsoil microbiome. *Nature* 560:233-237.

Reviewer #3 (Remarks to the Author):

This study collected root soil and root samples from 308 individual trees across a 3 220 km latitudinal gradient incorporating 18 locations between northern Norway and central Italy. Tree species were collected that host different plant symbionts (e.g. (*Alnus*: EcM/AM/N-fixing, *Betula*: EcM, and *Sorbus*:AM). These tree species co-occurred at most locations. It was tested whether symbiotic interactions contribute to shaping microbial communities and functions. Focus was on the soil microbiome. The manuscript is well written, presents an extensive data-set and contains some interesting observations, especially those related to dark septate endophytes (DSE).

Comments (in order of appearance).

Title: Enhanced potential of tree symbioses in shaping the soil microbiome under warmer climates. Is the title strongly supported by the presented data?

L28 across a European gradient. Perhaps add the size (3000 km) and mention the number of sites or tree individuals investigated to demonstrate the size of the study to the readers.

L38: probably also protists, nematodes and other soil biota.

L90: this is interesting. The correlation between root colonization and proportion of EcM fungal sequences is relatively weak (explained variance $R^2 = 0.15$). Is this often observed (e.g. for similar studies)? Or do they explain more (or less)?

L147: plant pathogens were significantly less abundant with DSE. Would it be possible to show this as a scatterplot (without the different DSE abundance categories). This sounds very interesting. Generally, it would be worthwhile to further explain which symbiosis types explain the occurrence of plant pathogens best (e.g. SEM including all three symbionts assessed as explaining variables). There is a lot of literature demonstrating EcM can suppress plant pathogens while this is less well observed for AMF. However, links between plant pathogens and DSE has rarely been assessed. This last aspect is interesting and should get some more attention.

The DSE results are most novel and interesting and it would be worthwhile to give this more attention (e.g. in the abstract).

L185: This study demonstrates that the effect of tree symbioses on soil microbial community properties and decomposition depends strongly on climate (Fig. 3a-c). However, Figure 3a-c only presents data for dual trees? Are the conclusions valid for all tree species (so including EcM and AM). Is there a way to integrate the results (including dual trees, EcM and AM trees into a coherent picture demonstrating the influence of climate on symbiosis and soil microbes in general (as the title suggests)?

L263: In this study we found little evidence to support the negative association of the EcM symbiosis with plant pathogens apart from the negative association of EcM conifer trees with the proportion of plant pathogens (Fig. 3b & Fig. S1), and we found that compared to AM roots, roots with low levels of EcM colonization were associated with higher proportions of plant pathogens in the soil (Fig. 3f). Reviewer: how does this look like when one puts the data together (e.g. without

the three abundance levels).

L310 Did you assess root colonization of *S. torminalis* or *S. domestica*

L414: Fungal OTUs were subsequently assigned to a primary lifestyle at the genus level using the FungalTraits tool [69]. How good is this data base to characterise DSE? How many DSE are in the fungal traits tool (would it be possible to add a table with the DSE taxa – based on sequences including those listed in the FungalTraits tool)? Please provide some more data. Where the roots that were heavily colonized by DSE sequenced (to identify the identity of the DSE)?

According to the methods, the root microbiome was not assessed. This is a pity as this would have provided further appealing information (e.g. links between root colonization and symbiont colonization, or symbiont diversity; links between the soil and root microbiome).

Why is DSE so strongly correlated with C/N ratio? This is interesting? Is EcM correlated with the C/N ratio? Any biological reason for the differences.

Are there any technical issues linked to this study (e.g. effects of primer choice on detected fungi, sampling intensity curves (number of OTUs versus sequencing depth) that are worth discussing?

Are there further data linked to this data-set (meta-genomics or the occurrence of functional genes, functional gene diversity). If so, are these data linked to the occurrence of symbiosis types (e.g. link to DSE)? If available, it would be worthwhile to include such data.

Fig 1: it would be possible to indicate the tree species found at each site to see where there is overlap and where not.

Fig 3: Would it be possible to show the actual strengths and significance levels of the different arrows (in addition to arrow thickness). Are the different levels of colonization defined in the methods – please describe carefully if not done-.

Figures 3 demonstrates Structural equation modelling showing the proposed direct and indirect drivers for specific groups focusing on dual-mycorrhizal trees. It would be worthwhile to also present such SEM models for other trees species (e.g. in the supplement).

Fig S4 demonstrates nicely that particular bacterial classes are linked to symbiosis types. Does this confirm other studies. Please provide a simple table (or an additional column with references if this has been found). For instance negative correlations with AMF abundance but also positive correlations with DSE (DA052) or EcM (e.g. Clostridia)

Reviewers' comments:

Reviewer #1 (Remarks to the Author):

Overall, this paper presents the results a large scale field sampling campaign. They leverage metabarcoding studies of bacteria and fungi to understand correlations between different mycorrhizal types, the soil microbiome and various climatological variables. This study is well conducted, but fails to meet the mark of novelty and significant insight that I have come to expect from Nature Communications.

We thank the reviewer for their comments and time taken for reviewing our manuscript. We have now substantially expanded our dataset to include both shotgun metagenomic data of functional genes for both soil and roots, as well as metabarcoding data of fungal and bacterial communities from the root microbiome. We believe these changes have greatly increased the significance and novelty of our insights, especially regarding the importance of dark septate endophytes.

Specific elements where this manuscript can be improved include delineation of the hypotheses. My understanding is that there is already decent evidence for Hypothesis 1, from global topsoil surveys. In particular hypotheses 2 and 3 are not aligned with the evidence provided by the introduction. For example, what does temperate mean in this case? It is poorly operationalized in this hypothesis and throughout the paper.

We have re-written the introduction and re-formulated our hypotheses to better match the insights gained from our study (L87-92). Specifically, we have focused on the effect of root colonization and removed our hypothesis about climate.

Positive elements of this paper include the work done to study the percent colonization of various symbionts and the overall methods, including the bioinformatics are state of the art. Besides the teabags decay experiment, I am missing a functional characterization of the soil communities that would aid in the interpretation and significance of the work.

We thank the reviewer for this comment. We have now expanded our dataset to include shotgun metagenomic data for functional gene characterization of soil and root microbiomes. Specifically, we have analyzed the effect of root symbiont colonization on the diversity, composition and proportions of different C, N, and P cycling genes, as well as general functional genes. The results are presented in lines 151-214, and Figs. 4 and 5. We revised the discussion accordingly in lines 288-340. This additional data facilitated additional novel insights on how dark septate endophytes may impact the functioning of belowground microbial communities.

Finally, the title of the paper is misleading, as it is unclear why warming would lead to any changes based on the evidence at hand. The title needs to be rewritten to encapsulate the main

findings of the study, which are limited in their capacity to predict future microbiome activity and function in forest landscapes

We agree with the reviewer. We have now changed the title to encapsulate the main findings of our (revised) study. The new title is: 'Pervasive effect of dark septate fungi on tree root and soil microbiomes'

Reviewer #2 (Remarks to the Author):

Netherway and colleagues examined how mycorrhizal associations affect surrounding soil microbiomes and ecosystem function across a latitudinal gradient in Europe. They found strong effects of EM and DSE fungi but smaller effects of AM fungi on the relative abundance of soil saprotrophs, pathogens and decomposition. In general, effect sizes were larger in temperate compared to boreal forests. This work is a nice overview of fungal effects belowground across climatic gradients and one of the first to included DSEs as well as AM and EM mycorrhizal fungi. However, I have several comments on the novelty, hypotheses, analysis, and conclusions of this work. Detailed comments are below.

Novelty: Several studies have targeted similar questions. Here it would be helpful to distinguish how the current work is different from previous studies on microbiomes associated with different mycorrhizal types (e.g., Bahram et al. 2018) or those focused on biogeochemical differences among mycorrhizal types.

We thank the reviewer for their comments. We have now greatly expanded our dataset and re-written the manuscript to emphasize how our study differs from previous studies on microbiomes associated with different mycorrhizal types. Specifically, this study is novel because:

- 1. This study is one of the very few studies that tackles how root symbionts (including mycorrhiza) affect microbiomes over a large environmental gradient using an extensive sampling campaign and considers the colonization rates of symbionts as opposed to just considering the binary of mycorrhizal type as an effect. Furthermore, these types of questions are usually addressed by studies on the local scale or meta-analyses that have their known drawbacks.*
- 2. We are unaware of any study that has looked at the effect of mycorrhizal colonization alongside DSE colonization on microbiomes.*
- 3. We believe this one of the most comprehensive studies of its kind: In addition to more traditional assessments of fungal root colonization (by microscopy) we assessed both soil and root fungal and bacterial communities and characterized their functional genes (N-cycling, P-cycling, CAZyme, total functional gene composition) using state-of-the-art molecular techniques. In addition, we included high-resolution climate data and soil quality and vegetation measurements to be able to better generalize our findings.*

4. *The main finding of this study is very novel, as we find strong evidence for an unexpectedly large role of DSE in influencing belowground communities and their potential functioning in natural ecosystems across a large environmental gradient.*

We have now emphasized the novelty of our work on lines 217-229 and 366-389.

Hypotheses: Overall the hypotheses are difficult to follow in the results and discussion sections. It would be helpful to include language in both sections about testing certain hypotheses and if they were rejected.

We thank the reviewer for their comment. We have re-written our hypotheses which are now as follows:

'We hypothesized that both mycorrhizal and DSE colonization affect soil and root microbiome structure and potential functions, independently of climate and soil properties. More specifically we expected that – as they often monopolize roots – colonization by mycorrhiza and DSE suppresses plant pathogens, while also altering the proportion of taxa and genes involved in decomposition and nutrient cycling.'

We present and discuss the results now in a more logical order following from the hypotheses and reflect on our findings in relation to the proposed hypotheses throughout the discussion.

In addition, the hypotheses could be clearer by pointing out the contrasting conditions you've studied. For example, why would temperate forests have the highest impact? Compared to what other ecosystem(s)?

After the addition of the large new datasets, the focus of our study has shifted somewhat. We have rewritten the hypotheses and have removed explicit hypotheses on how we expect the contrasting environmental conditions would affect root symbiont associations with the soil microbiome. The new hypotheses can be found in the previous response and L87-92 in the manuscript.

Is colonization reflective of the strength of symbiosis? What if the colonizing fungi are more parasitic than mutualistic?

We have removed mentions of colonization reflecting the strength of the symbiosis, and re-worded colonization as reflecting the prevalence of the symbiosis. We agree with the reviewer that the colonizing fungi may act on a mutualism-parasitism spectrum, yet most evidence on EcM, AM, and DSE suggests that their colonization is generally associated with positive effects for the host plants, and we cannot explicitly test where they are on a mutualism-parasitism spectrum.

Statistics. It is unclear how the linear models accounted for the hierarchical design of the plots and sites. Each model should have plot within site as a random effect. Do the Spearman rank correlations include this information?

We have re-done the analyses to account for the hierarchical study design.

- We have now performed all analyses on the level of group means or composite sample, i.e. mean or composite sample per species per site/plot, this accounts for the non-independence of individual trees from the same species occurring in the same site/plot, and brings our data onto the same level

- We have fitted mixed-effects models on our mean/composite values with plot within site as a random effect

- For all SEM models presented we have also used mixed-effects models with plot within site as a random effect

- We could not include random effects in the Spearman correlation and random forest analyses, yet we used these methods for data exploration and for variable selection, and for specific relationships of interest we implemented mixed models as explained above.

- For the new PERMANOVA analysis of community/gene compositions we could not include a random effect as strata due to the unbalanced design, but we included Site as a fixed effect to at least account for some of its possible influence

Colonization v. molecular relative abundance: It's unclear why both methods were performed. In addition, how was the relative abundance of the different fungal groups calculated in the soil? Are the databases good enough to define DSEs accurately? How many taxa were assigned to each group? If most of the sequences were assigned to a fungal functional group, then relationships in each focal group could be caused by the change in another group, not necessarily the focal group.

References

Bahram et al. 2018. Structure and function of the global topsoil microbiome. *Nature* 560:233-237.

We thank the reviewer for the comment. While we agree to some extent, we still believe that including both measures of symbiont prevalence is informative as they measure different aspects of the symbioses, we also believe that despite their shortcomings using molecular relative abundances of guilds/taxa contains useful information.

- We used both colonization and molecular relative abundances to verify one another and avoid using molecular relative abundances to explain the relative abundances of other groups from the same molecular dataset. Colonization is independent of the constraints imposed by the compositionality of molecular data (e.g., such as the performance of the databases, working with relative abundances rather than absolute abundances etc.). We also think these are useful data for the research community as we are unaware of studies that have compared the methods for multiple symbionts across a large environmental gradient.

- We calculated the relative abundance of different fungal and bacterial groups using the centered-log ratio transformation, while not perfect, this method acknowledges the compositionality of molecular data and retains both absolute and relative abundance information by comparing groups to the geometric mean for each sample. For discussions of this matter see Gloor, et al., *Microbiome Datasets Are Compositional: And This Is Not Optional*. *Frontiers in Microbiology*, 2017. **8**.

- The database we used (*Fungal Traits*) categorizes guilds at the genus level. Within this database potential DSE genera have been listed based on the literature at the point of database formation and curated by experts. We have further curated the potential DSE taxa on roots and in soil by removing genera with known ectomycorrhizal and ericoid mycorrhizal taxa and we have provided this list of potential DSE taxa and their sequences as supplementary tables. We agree that this is imperfect, and as it stands it is an extremely slow process to designate fungi as DSE, as this requires isolating pure cultures from roots and then re-inoculating roots and confirming the specific strain forms DSE structures. Therefore, we have mentioned that our molecular characterization of DSE only represents potential DSE at best, yet we have performed colonization analysis of observed DSE structures in roots and this has correlated quite strongly with our molecular characterization of potential DSE (see new Figure 2i).

- We agree that like any metabarcoding study, our microbiome data are compositional and that relationships in each focal group could be caused by the change in another group, not necessarily the focal group; indeed, as one relative abundance of a particular group changes the relative abundance of other groups must simultaneously change. This is the nature of working with metabarcoding data and we note that other methods of analyzing groups of microbes are equally riddled with their own issues. Furthermore, qPCR analysis of specific groups has been shown to correlate with relative abundances of these groups from metabarcoding or metagenomics (e.g., Bahram et al. 2022 *Nat Communications*), and we have shown here that root colonization (determined by microscopy) has also correlated with the molecular relative abundance of certain groups. Also, our observations, e.g., the negative correlation of DSE with potential plant pathogens, is backed up by experimental studies see *Discussion (L 231-263)*

Reviewer #3 (Remarks to the Author):

This study collected root soil and root samples from 308 individual trees across a 3 220 km latitudinal gradient incorporating 18 locations between northern Norway and central Italy. Tree

species were collected that host different plant symbionts (e.g. (Alnus: EcM/AM/N-fixing, Betula: EcM, and Sorbus:AM). These tree species co-occurred at most locations. It was tested whether symbiotic interactions contribute to shaping microbial communities and functions. Focus was on the soil microbiome. The manuscript is well written, presents an extensive data-set and contains some interesting observations, especially those related to dark septate endophytes (DSE).

Comments (in order of appearance).

Title: Enhanced potential of tree symbioses in shaping the soil microbiome under warmer climates. Is the title strongly supported by the presented data?

We thank the reviewer for their comments. We have changed the title to reflect our presented data, which is greatly expanded from the previous version. The new title is: 'Pervasive effect of dark septate fungi on tree root and soil microbiomes'

L28 across a European gradient. Perhaps add the size (3000 km) and mention the number of sites or tree individuals investigated to demonstrate the size of the study to the readers.

We have added this information now (L 81-83)

L38: probably also protists, nematodes and other soil biota.

We agree and mention 'mainly bacteria and fungi' (L 47)

L90: this is interesting. The correlation between root colonization and proportion of EcM fungal sequences is relatively weak (explained variance $R^2 = 0.15$). Is this often observed (e.g. for similar studies)? Or do they explain more (or less)?

We have now examined the molecular proportion of EcM fungi on roots (previously soil) and did not find a significant relationship with colonization by microscopy, although when solely looking at Betula and Alnus incana roots there is a significant and quite strong relationship. We could not find any studies that have compared these two methods before in such a way. As we extracted DNA from root systems and not just root tips which is often done for EcM, this may account for the weak relationship, i.e., if the proportion of EcM fungi was from root tips only and from an equal amount of root tips that were examined under the microscope then there may indeed be a stronger relationship between the two methods.

L147: plant pathogens were significant less abundant with DSE. Would it be possible to show this as a scatterplot (without the different DSE abundance categories). This sounds very interesting. Generally, it would be worthwhile to further explain which symbiosis types explain the occurrence of plant pathogens best (e.g. SEM including all three symbionts assessed as explaining variables). There is a lot of literature demonstrating EcM can suppress plant pathogens while this is less well observed for AMF. However, links between plant pathogens and DSE has rarely been assessed. This last aspect is interesting and should get some more attention.

We thank the reviewer for their interest in our results. We have now presented scatterplots showing that DSE colonization is negatively correlated with the proportion of putative plant pathogens in both roots and soil, and we have also simultaneously presented SEM analyses that show that these relationships hold up when including environmental variables (Fig. 3). We have in general made the observations concerning DSE central to our manuscript and discuss the potential mechanisms and cited previous literature on the matter extensively in the discussion.

The DSE results are most novel and interesting and it would be worthwhile to give this more attention (e.g. in the abstract).

We agree, and our additional data has confirmed this, thus the DSE results are now emphasized more throughout the manuscript

L185: This study demonstrates that the effect of tree symbioses on soil microbial community properties and decomposition depends strongly on climate (Fig. 3a-c). However, Figure 3a-c only presents data for dual trees? Are the conclusions valid for all trees species (so including EcM and AM). Is there a way to integrate the results (including dual trees, EcM and AM trees into a coherent picture demonstrating the influence of climate on symbiosis and soil microbes in general (as the title suggests)?

We have changed our manuscript and title to more reflect the main findings of our study and have moved away from our focus on climate. After reanalysis of existing data and the newly added data, we now show that the effect of climate is less strong compared to soil factors such as pH and CN-ratio. Also root colonization appears to be nearly as important as climate for explaining many microbiome properties (Fig. 5). We now present figures showing all trees and not just dual mycorrhizal trees (see Figs 3 and 4).

L263: In this study we found little evidence to support the negative association of the EcM symbiosis with plant pathogens apart from the negative association of EcM conifer trees with the proportion of plant pathogens (Fig. 3b & Fig. S1), and we found that compared to AM roots, roots with low levels of EcM colonization were associated with higher proportions of plant pathogens in the soil (Fig. 3f). Reviewer: how does this look like when one puts the data together (e.g. without the three abundance levels).

We have now present scatterplots showing these relationships without the abundance levels (please see Fig. 3)

L310 Did you assess root colonization of *S. torminalis* or *S. domestica*

*Yes, indeed we assessed root colonization on both *S. torminalis* or *S. domestica*, We have now made this clear by giving these species a different color in the scatterplots. They generally fit into the same patterns shown for *Sorbus aucuparia*.*

L414: Fungal OTUs were subsequently assigned to a primary lifestyle at the genus level using the FungalTraits tool [69]. How good is this data base to characterise DSE? How many DSE are in the fungal traits tool (would it be possible to add a table with the DSE taxa – based on sequences including those listed in the FungalTraits tool)? Please provide some more data. Where the roots that were heavily colonized by DSE sequenced (to identify the identity of the DSE)?

We thank the reviewer for the comment. The Fungal Traits database categorizes guilds at the genus level, within this database DSE is not a primary lifestyle assignment, instead there is a category of classification called endophytic capacity, within which potential DSE genera have been listed based on the literature at the point of database formation and curated by experts. We have further curated the potential DSE taxa on roots and in soil by removing genera with known ectomycorrhizal and ericoid mycorrhizal taxa and we have provided this list of potential DSE taxa and their sequences as supplementary tables. As DSE status potentially changes at the strain level, we can at best assume that our molecular characterization of DSE only represents potential DSE genera. Yet we have performed colonization analysis of observed DSE structures in roots and this has correlated quite strongly with our molecular characterization of potential DSE in both roots (new data) and soil see Figs. 2 and S1. We have now also performed analysis of specific OTUs, and genera most strongly associated with DSE colonization, we found two OTUs that correlated strongly with colonization, we also found specific genera the correlated strongly with DSE colonization, we have presented these data on lines 204-209, Fig. S10.

According to the methods, the root microbiome was not assessed. This is a pity as this would have provided further appealing information (e.g. links between root colonization and symbiont colonization, or symbiont diversity; links between the soil and root microbiome).

We thank the reviewer for the suggestion. We have now added root microbiome data, and show that it mostly reinforces findings from the soil microbiome see Figs. 3, S1, S3, and S4,

Why is DSE so strongly correlated with C/N ratio? This is interesting? Is EcM correlated with the C/N ratio? Any biological reason for the differences.

We have now discussed this link in the discussion see L 315-317 and 327-329. It is difficult to tease apart cause and effect here, but DSE colonization has previously been associated with higher soil organic matter content, and also DSE are generally more abundant in cold and infertile environments, in accordance we found a negative association between DSE colonization and MAT, and soil C/N was negatively associated with MAT, MAP and pH (see Fig. S1). While it possible that DSE are strongly associated with C/N as they are potential saprotrophs as well, they may also contribute to higher C/N ratios through their contribution to more recalcitrant organic matter by adding highly melanized necromass. For EcM fungi both molecular and microscopy abundances showed weak correlation with C/N and were most tightly

associated with soil moisture and climatic moisture deficit, this is not completely out of line with existing evidence, that suggests EcM associations are sensitive to moisture availability see L 219-220.

Are there any technical issues linked to this study (e.g. effects of primer choice on detected fungi, sampling intensity curves (number of OTUs versus sequencing depth) that are worth discussing?

To account for differences in sampling depth we rarefied our data for analysis of alpha diversity (Shannon H index) to make it comparable across our study. We did not look at OTU richness as a response variable because it is more sensitive to sequencing depth compared to Shannon H index, and while we took a conservative approach to analyzing diversity, it is at least comparable across our study. We discuss potential limitations of using metabarcoding data see L 360-364. While our primers used for fungi (ITS2) are not ideal for detecting AM fungi, we found a positive correlation between microscopic AM colonization and the proportion of AM ITS2 sequences (Fig. 2f).

Are there further data linked to this data-set (meta-genomics or the occurrence of functional genes, functional gene diversity). If so, are these data linked to the occurrence of symbiosis types (e.g. link to DSE)? If available, it would be worthwhile to include such data.

In addition to adding root metabarcoding data of fungi and bacteria, we included shotgun metagenomic analysis of functional genes for both soil and roots. This has greatly expanded the dataset and strengthened the findings of our study. We have further linked specific CAZyme genes and general functional genes to DSE colonization (L 202-214).

Fig 1: it would be possible to indicate the tree species found at each site to see where there is overlap and where not.

Done, this information is now added to Figure 1.

Fig 3: Would it be possible to show the actual strengths and significance levels of the different arrows (in addition to arrow thickness). Are the different levels of colonization defined in the methods – please describe carefully if not done-.

All our SEMs now include strengths and significant levels within the figures. We have removed the different levels of colonization from analysis and instead show scatter plots of the analysis, similar patterns are shown.

Figures 3 demonstrates Structural equation modelling showing the proposed direct and indirect drivers for specific groups focusing on dual-mycorrhizal trees. It would be worthwhile to also present such SEM models for other tree species (e.g. in the supplement).

We could only analyze AM and DSE colonization from Alnus and Sorbus within the same SEM, we had to exclude Betula as that only had EcM colonization analysis performed, and SEM/linear mixed models do not allow for unbalanced designs. And as EcM colonization did not appear to have strong effects in the linear mixed models we did not follow up with EcM effects in SEM.

Fig S4 demonstrates nicely that particular bacterial classes are linked to symbiosis types. Does this confirm other studies. Please provide a simple table (or an additional column with references if this has been found). For instance negative correlations with AMF abundance but also positive correlations with DSE (DA052) or EcM (e.g. Clostridia)

We also think the association between different bacterial taxa and root colonization is interesting, yet we have removed this figure and analysis from our manuscript, and it has been replaced by new analysis and figure (see new Fig. S4). This new analysis and figure look at both root and soil bacteria. Particularly in roots we found many associations between DSE and bacterial phyla/classes, and few with EcM and AM, we outline these results briefly in the Supplementary information.

REVIEWER COMMENTS

Reviewer #1 (Remarks to the Author):

Overall, I have significant concerns primarily about the designation of DSE as a group for analysis, as well as the justification of the hypotheses. Both points can potentially be fixed in revisions, but they are substantial, requiring rewriting and reanalyzing substantial portions of the datasets and may not be of interest for the authors. Overall, the methodology used is cutting-edge and reflects a substantial improvement over the previous iteration of the ms. The gradient style approach utilizing multiple tree hosts is of interest, and allows for some level of generality about the abundance of root-endophytes and their relative impact on root/soil microbiomes

Overall:

Dark septate endophytes are an ambiguous group of fungi, with poorly defined taxonomy. Meaning that whether an Ascomycete fungi detected in a root, is or is not, a DSE is poorly understood. The typical definition of DSE reflects this ambiguity : "Fungi that colonize living plant roots by melanized septate hyphae" Ruotsalainen et al., 2022 TREE. Endophytic root-fungi are highly diverse, many of which form close associations with EM roots (Toju & Sato 2018 Frontiers Microbio; Pellitier et al., 2021 Fungal Ecology). The functioning of these highly cryptic ascomycetous fungi is poorly known with many having dual roles in databases such as FungalTraits as pathogens/saprotrophs/endophytes. I think that exploring the role of root endophytes is very important, but am concerned that not enough information on DSE is known to make the sweeping claims presented in this ms. As it stands, our ability to characterize DSE from sequence data is poor. Calling them Potential DSE is helpful, but care must be taken to explain the challenges inherent with characterizing DSE to start with, given that it is purely a morphologically defined definition. The inclusion of the Table S1, is also helpful, however, only in few cases are the morphological analyses attached to species names or even genera. My suggestion is to reanalyze the data using similar analyses but with all endophytic root fungi, and make a special discussion section for the potential role of DSE in driving the broader endophyte patterns observed here. Focusing on all root endophytes will have broader appeal, and likely broader impact.

Methods:

How were DSE fungi identified? Presumably they were identified using the taxonomic identify at the genus level. My concern however is about the fact that many fungi in FungalTraits are not known to be exclusively DSE, many are identified as endophytes, pathogens and or DSE. In the introduction, and here, much more stringent parameters needed to be provided to first describe the biology and phylogenetic breadth of DSE, as well as how they were identified in FungalTraits when multiple entries for lifestyle were given.

Alnus roots host N fixing bacteria, please clarify if nodules were accounted for or measured and how they were incorporated into the analyses. The measurement solely of N fixing genes seems insufficient to account for confounding effects of these nodules on overall community structure. Without measuring the proportion of these important symbionts, I am not clear on what soil chemical analyses can be effectively utilized as predictors.

Please clarify how AM vs DSE were independently measured using the grid-intersection method. They both stain nearly identically with Trypan, so it is not clear what morphological character was used to distinguish them.

How was host identity utilized in the LME? Was it included an effect? If so how? The challenge being is that most of the tree hosts used are not distributed across the breadth of the gradients presented.

Figures:

2c in particular is not clear what is being measured on the y axis. How can the proportion be larger than 100? Please relabel or replot the data to make this more clear, both here and elsewhere where the clr is utilized. The above comment applies to figure 3 as well.

Figure 4, the potential for N-fixing bacteria in alnus to be impacting the results presented in panel j,k,l, as well as p,q,r, needs to be considered.

Results/Discussion:

I am concerned about the overall justification of the hypotheses. While I agree that endophytic or mycorrhizal fungi could influence the soil or root microbiome, I would argue that climate plays a significant role in shaping the distribution of AM or EM symbiose (sensu Steidinger et al., 2019 Nature, or van der Linde 2018 Nature). Accordingly the presentation of the results as being able to derive variance partitioned between climate and EM/AM/DSE feels misleading. I feel a key analysis missing is exploring climate/soil variable explaining the occurrence and abundance of DSE/EM/or AM fungi. No presentation of the latent impacts of climate in driving the distribution of these major AM/EM or DSE is presented, leading to misleading claims about the general role of these fungi in forest systems. I understand that such perspectives run somewhat counter to those presented by Phillips et al., 2013, and the MANE hypothesis, but the role of climate in mediating the distribution of key symbioses cannot be ignored.

Similar to the above point, the key drivers of soil fungi, root fungi, and most other measured variables are primarily attributable to soil and climatic variables, not the presence or AM/EM or DSE. i.e. Figure 5, to this point, Section starting at line 267 needs to present a much more nuanced picture of the relative importance of climate, alongside DSE (or endophytes more broadly). As written this section is misleading when compared against Figure 5a,b.

Line 222, why do you think this is the case here, when it has been widely found to be the opposite in many studies.

Reviewer #3 (Remarks to the Author):

The manuscript has much improved and I really like it now. Earlier I was not so convinced that it was strong enough for Nature Communications (in that form), but now it includes many appealing data and the role of dark septate endophytes is much clearer shown now. I have a few minor comments:

Line 2: title suggestion: dark septate fungi drive tree root and soil microbiome composition and function in European forests.

- if you would choose this title (and the editor agrees), obviously it needs to be made clear in the discussion that further field and experimental studies are needed to verify the observations made here (especially the important role of dark septate endophytes).-

Abstract: it would be worthwhile to mention the effects of dark septate endophytes (DSE) on pathogens (e.g. linked to the results presented in figure 3). Generally, the effects of DSE could even be more strongly emphasized in the abstract, also specifically mentioning their effects on a range of parameters including effects on pathogens, on fungal diversity, on functional gene composition etc. (to "visualize" the strong effects to the reader).

L97-101: very long sentence. Consider to have two sentences. In general, the manuscript could benefit a little (minor issue, no major issue) if a native speaker goes over it (this reviewer is also not a native speaker).

The discussion on the importance of DSE and their role in the soil is very useful (including potential mechanistic insights of how DSE influence pathogens and microbiome composition).

L446: the description of DSE detection (microscopy) is very short. Please add a few pictures (supplement) to show how DSE look like and explain the morphological characteristics used to count a DSE hyphae in more detail.

Reviewer #4 (Remarks to the Author):

The authors present a thorough contribution detailing the potential effects of EcM, AM, and DSE on soil and root microbiomes and functional potentials. While I was charged by the editor to address one specific aspect of this work, namely the efficacy of previous revisions in responding to previous reviewers, I do also include a few other concerns here.

Overall, the authors present interesting results.

Lines 48-50: I do not think that this is a fair statement. There have been many previous studies that have explored how root symbionts impacts below ground studies, even with differential mycorrhizal status. This statement suggests a poor understanding of literature from the last 20 years.

Lines 105-106: Why do you think that EcM does not correlated with EcM here? And do you think that this lack of correlation where there should be one is concerning? How does this reflect of the rest of the data analyses?

Lines 168-178: DSE can be hard to find and can be prone to false negatives. That is, if you have a zero for colonization, how confident are you that it is a true zero? But these zeros are included in all of these analyses. It would be nice to omit these zeros and re-run these statistics to confirm that these general patterns still hold without a plethora of zero counts potentially artificially pulling the slope down,

180-185: I would suggest omitting this (and in methods) as it adds little to the story.

My biggest concern is that plant species was not controlled for. You have the categories of AM, EcM and DSE across a few species within genera. But individual species are not controlled from in analyses. With the exception of a recent paper in Fagaceae, almost all literature suggests that species (really, host genetics) play the predominant role in structuring mycorrhizal and soil communities. It is possible that all patterns seen here are species effects, rather than symbiont effects. This must be accounted for and discussed.

REVIEWER COMMENTS

Reviewer #1 (Remarks to the Author):

Overall, I have significant concerns primarily about the designation of DSE as a group for analysis, as well as the justification of the hypotheses. Both points can potentially be fixed in revisions, but they are substantial, requiring rewriting and reanalyzing substantial portions of the datasets and may not be of interest for the authors. Overall, the methodology used is cutting-edge and reflects a substantial improvement over the previous iteration of the ms. The gradient style approach utilizing multiple tree hosts is of interest, and allows for some level of generality about the abundance of root-endophytes and their relative impact on root/soil microbiomes

Response: We thank the reviewer for their in-depth and constructive review of our manuscript.

Overall:

Dark septate endophytes are an ambiguous group of fungi, with poorly defined taxonomy. Meaning that whether an Ascomycete fungi detected in a root, is or is not, a DSE is poorly understood. The typical definition of DSE reflects this ambiguity : “Fungi that colonize living plant roots by melanized septate hyphae” Ruotsalainen et al., 2022 TREE. Endophytic root-fungi are highly diverse, many of which form close associations with EM roots (Toju & Sato 2018 Frontiers Microbio; Pellitier et al., 2021 Fungal Ecology). The functioning of these highly cryptic ascomycetous fungi is poorly known with many having dual roles in databases such as FungalTraits as pathogens/saprotrophs/endophytes. I think that exploring the role of root endophytes is very important, but am concerned that not enough information on DSE is known to make the sweeping claims presented in this ms. As it stands, our ability to characterize DSE from sequence data is poor. Calling them Potential DSE is helpful, but care must be taken to explain the challenges inherent with characterizing DSE to start with, given that it is purely a morphologically defined definition. The inclusion of the Table S1, is also helpful, however, only in few cases are the morphological analyses attached to species names or even genera. My suggestion is to reanalyze the data using similar analyses but with all endophytic root fungi, and make a special discussion section for the potential role of DSE in driving the broader endophyte patterns observed here. Focusing on all root endophytes will have broader appeal, and likely broader impact.

Response: We thank the reviewer for these comments about DSE. We agree that DSE are an ambiguous group of fungi with poorly defined taxonomy and purely a morphological definition (this is now noted in the introduction and discussion on L67-72 and L449-457.

- We note that the molecular characterization of potential DSE fungi presented in our manuscript plays only a small part in the analysis - it was merely used to see if there was a taxonomic/molecular data signal from our sequencing data that correlated with our

microscopic quantification of DSE colonization, and thus it has no repercussions for any other analysis.

- It is correct that DSE is not a primary lifestyle in the FungalTraits database and exists as an additional supplementary categorization under 'root endophytic capacity'. In addition, indeed all genera we included for our calculations of potential DSE fungi have a primary lifestyle assignment as e.g., saprotrophs and endophytes, and our calculation of the relative abundance of potential DSE fungi was performed separately from the primary lifestyle assignments, i.e., our potential DSE relative abundances do not affect the relative abundance calculations of other guilds and are thus supplementary. We have now provided a clearer explanation of how we calculated relative abundances of potential DSE fungi in the methods section (L646-650 & 737-740).
- We also note that DSE colonization is poorly correlated with the relative abundance of fungal genera with the primary lifestyle assignment of root endophyte in our root samples (see Figure Aa below). DSE colonization was also not significantly correlated with the relative abundance of all fungi with a potential root endophytic capacity, i.e., all possible endophytes (Figure Ab), compared to a significant and stronger correlation between the relative abundance of fungi we designated as potential DSE taxa in roots (see Fig.2h, $R^2_m = 0.20$, $p = 3e-03$). We have now presented this in the results L136-140.
- We now highlight the need for a better understanding of the fungi currently grouped as DSE. Therefore, we explicitly advocate for studies gaining us better understanding of the DSE lifestyle and how they affect plants, soil processes and other organisms etc. in L452-467.

Although we agree that root endophytes in general may have a broader appeal so we have included mentions of root endophytes in general as much as possible throughout the manuscript including in the introduction, discussion, and implications sections (L68, 309, 336, 423, 488, & 495); yet, our analysis and major findings are related specifically to the quantification of dark-septate endophyte structures in roots via microscopy (DSE colonization), and therefore we think DSE deserve to be a major focus in the manuscript and appear to have a unique effect in our dataset that cannot be generalized to all root endophytes.

Figure A: Results from linear-mixed effects model showing a) the relationship between the relative abundance (centered-log ratio) of fungi on roots with the primary lifestyle classification as root endophytes and DSE colonization, and b) the relationship between the relative abundance (centered-log ratio) of all fungi on roots with a potential root endophytic capacity.

Methods:

How were DSE fungi identified? Presumably they were identified using the taxonomic identify at the genus level. My concern however is about the fact that many fungi in FungalTraits are not known to be exclusively DSE, many are identified as endophytes, pathogens and or DSE. In the introduction, and here, much more stringent parameters needed to be provided to first describe the biology and phylogenetic breadth of DSE, as well as how they were identified in FungalTraits when multiple entries for lifestyle were given.

Response: Please see our response to the previous comment above, we have added information on how we assigned fungi and as potential DSE on L646-650 & L737-740, and discussed their ambiguity on L67-72 and L449-457.

Alnus roots host N fixing bacteria, please clarify if nodules were accounted for or measured and how they were incorporated into the analyses. The measurement solely of N fixing genes seems insufficient to account for confounding effects of these nodules on overall community structure. Without measuring the proportion of these important symbionts, I am not clear on what soil chemical analyses can be effectively utilized as predictors.

Response: We thank the reviewer for making this important point. We did not account for nodules nor measured them, yet we could obtain the relative abundance (clr transformed 16S reads) of the nodule forming bacterial genus *Frankia* on *Alnus* roots. We found the relative

abundance of *Frankia* on *Alnus* roots was positively correlated with soil pH ($R^2_m = 0.28$, $p = 3e-03$) out of abiotic factors, and out of microbiome properties positively correlated with the diversity of N cycling genes in roots ($R^2_m = 0.33$, $p = 4e-03$) and soil ($R^2_m = 0.18$, $p = 0.03$), total bacterial functional gene diversity in roots ($R^2_m = 0.28$, $p = 7e-03$), the ratio of bacteria/fungi in soil ($R^2_m = 0.25$, $p = 0.03$), and the relative abundance of N fixing genes in roots ($R^2_m = 0.45$, $p = 4e-04$). These results are now included in the results section (L142-148) and Fig. S1. This is now discussed on L413-424 in the discussion. We further address the concern of N fixing bacteria effects in the specific comment about the figures on the diversity of N cycling genes below.

Please clarify how AM vs DSE were independently measured using the grid-intersection method. They both stain nearly identically with Trypan, so it is not clear what morphological character was used to distinguish them.

Response: AM structures clearly stain blue with trypan, whereas DSE structures retain their dark pigmentation, both AM and DSE colonization are commonly co-quantified using trypan and the grid line intersection method (See Kohout *et al.* 2012 *FEMS Microbiology Ecology*, Massenssini *et al.* 2014 *Mycorrhiza*, Sudová *et al.* 2018 *AJB*, Muthukumar *et al.* 2006 *Mycorrhiza*, Uma *et al.* 2012 *Journal of Forestry Research*). We have now extended our methods description of DSE and AM quantification and added photos of DSE structures from our samples in the supplementary information - “*The root colonization rate (% root length colonized) of AM fungi and DSE was assessed on Alnus and Sorbus fine roots by the grid line intersection method [68]. First, 1 cm pieces of cleaned roots (around 10 cm total) from each tree sample were cleared in 2.5% KOH solution and stained with trypan blue in acidic glycerol solution [69]. Stained roots were then mounted on slides, and colonization was estimated under a minimum of 100 random fields of intersection by using a ZEISS Axioscope 5 Digital Microscope (Carl Zeiss Microscopy, Oberkochen, Germany) at X 400 magnification. For AM fungi the presence of stained arbuscles, hyphae, and vesicles (if present) were counted separately, and DSE were counted when darkly pigmented, septate hyphae and clusters of inflated, rounded, and thick-walled cells (microsclerotia) were present within the cortical cells of roots [70] (for photographic examples of DSE structures in our root samples see Fig. S8).*”

How was host identity utilized in the LME? Was it included an effect? If so how? The challenge being is that most of the tree hosts used are not distributed across the breadth of the gradients presented.

Response: We have now re-run all LME analysis to include host identity as part of a crossed random effect structure with Site/plot e.g. + (1| Site:Plot) + (1| Tree_species), whereas previously we only had Site/plot as a random effect. The new analysis has slightly changed the results, yet Site:Plot was generally more important than host identity, and host identity often explained very little variation; we have now included all model outputs in the supplementary material.

Figures:

2c in particular is not clear what is being measured on the y axis. How can the proportion be larger than 100? Please relabel or replot the data to make this more clear, both here and elsewhere where the clr is utilized. The above comment applies to figure 3 as well.

Response: We have now changed all mentions of 'proportions' in the manuscript and figures to 'relative abundances' to avoid confusion, and clearly labeled the axes.

Figure 4, the potential for N-fixing bacteria in alnus to be impacting the results presented in panel j,k,l, as well as p,q,r, needs to be considered.

Response: The relationship between DSE colonization and N cycling gene diversity in soil was no longer significant when accounting for the random effect of tree species in the LME model (now moved to Fig. S3p), however the stronger negative relationship between DSE colonization and the diversity of N cycling genes in roots (Fig. 3m) was robust to

- i) the random effect of tree species
- ii) was significant in the same LME model as the relative abundance of *Frankia* on roots e.g.
Diversity N cycling genes roots ~ Frankia + DSE colonization
Where $p = 0.01$ (*Frankia*), $p < 0.01$ (DSE), $R^2_m = 0.55$, $AIC = -37.74$
- iii) Was a better fitting model to explain the diversity of N cycling genes in roots than *Frankia* alone (*Frankia*: $R^2_m = 0.24$, $p = 0.001$, $AIC = -35.22$; vs DSE: $R^2_m = 0.37$, $p = < 0.001$, $AIC = -37.17$)
- iv) Was robust in a SEM model (DSE: std. coeff = -0.37 , $p = 2e-03$) that included MAT (std. coeff = -0.48 , $p = 4e-04$), pH (std. coeff = 0.32 , $p = 2e-03$), CN (std. coeff = -0.31 , $p = 0.02$), and the relative abundance of *Frankia* (std. coeff = 0.27 , $p = 6e-03$).
- v) There was not a significant relationship between DSE colonization and the relative abundance of *Frankia* ($R^2_m = 0.02$, $p = 0.39$)

These results indicate that the effect of DSE on N cycling genes is generally robust to the effect of *Frankia*, and these results are now presented in the results section L142-148, discussion L413-424, Figs. S2c-h, and Table S3).

Results/Discussion:

I am concerned about the overall justification of the hypotheses. While I agree that endophytic or mycorrhizal fungi could influence the soil or root microbiome, I would argue that climate plays a significant role in shaping the distribution of AM or EM symbiose (sensu Steidinger et al., 2019 Nature, or van der Linde 2018 Nature). Accordingly the presentation of the results as being able to derive variance partitioned between climate and EM/AM/DSE feels misleading. I feel a key analysis missing is exploring climate/soil variable explaining the occurrence and abundance of DSE/EM/or AM fungi. No presentation of the latent impacts of climate in driving the distribution of these major AM/EM or DSE is presented, leading to misleading claims about the general role of these fungi in forest systems. I understand that such perspectives run somewhat counter to those presented by Phillips et al., 2013, and the MANE hypothesis, but the role of climate in mediating the distribution of key symbioses cannot be ignored.

Response: We note that in our hypotheses, the line “*We hypothesized that both mycorrhizal and DSE fungal colonization would affect soil and root microbiome structure and potential functions independently of climate and soil properties*” was perhaps poorly worded. We did not mean to imply that root symbioses were more important than climate or soil properties or were in fact independent of these properties, so we have changed ‘*independent of*’ to ‘*in addition to and mediated by climate and soil properties*’ (now LXX). What we meant to imply was that root symbioses would have a detectable effect on the soil microbiome after accounting for climate and soil properties, and we hope the new wording reflects this.

We agree that climate and the environment in general plays a significant role in shaping the distribution of AM or EM symbioses, especially when looking at the mycorrhizal type distribution of host trees as shown in Steidinger et al., 2019 Nature, and the distribution and diversity of the symbionts as shown in van der Linde 2018 Nature. In this manuscript, we are bringing a novel perspective by quantifying the actual presence of different root symbioses in the form of root colonization rates and relating them to the soil and root microbiome while encapsulating the large climatic and edaphic range of some widespread tree species across Europe.

- i) We note that we have first established the main environmental drivers of the extent of root colonization by different symbionts (Fig. 2 & Fig. S1 & L101-104),
- ii) we have shown the robustness of the major relationships between DSE colonization and microbiome properties after accounting for soil and climate factors using SEM (Fig. 4),
- iii) established that DSE (and EcM in some cases) explains significant variation in microbial communities and gene compositions. Our PERMANOVA analysis (Fig. 5) that also included climate and soil factors (in some cases explaining more variation than these factors, and
- iv) we have added (Fig. 6) based on variation partitioning within sites, that shows that the size of the effect of both soil properties and root symbioses on root and soil microbiomes is dependent on latitude (as a proxy for climatic and environmental gradients) and the effects of both root symbioses and soil properties are generally enhanced in the middle of our latitudinal gradient.

Similar to the above point, the key drivers of soil fungi, root fungi, and most other measured variables are primarily attributable to soil and climatic variables, not the presence or AM/EM or DSE. i.e. Figure 5, to this point, Section starting at line 267 needs to present a much more nuanced picture of the relative importance of climate, alongside DSE (or endophytes more broadly). As written this section is misleading when compared against Figure 5a,b.

Response: Please see the above response in regards to how we addressed the effect of root symbioses in addition to and taking into account the effect of climate and soil properties. In the discussion we have tried to present a more nuanced picture by now alluding to the (oftentimes) stronger effect of climate and soil compared to root symbioses see L297-300, L340-342, L345-348, L351-354, L364-369, L435-447, & L491-497.

Line 222, why do you think this is the case here, when it has been widely found to be the opposite in many studies.

Response: Most studies addressing the effect of mycorrhizal associations on microbial communities are done so from a plant mycorrhizal type perspective, i.e., a binary plant trait that does not take into account the actual presence or prevalence of the mycorrhiza, or use molecular based relative abundances to quantify symbionts, to our knowledge there has not been a broadscale study addressing the effect of mycorrhizal root colonization rates on microbiome properties, which may explain our results. That being said, the effects of specific EcM guilds or taxa may be stronger compared to considering EcM as a unified guild, as EcM are not a functionally homogenous group (either?), AM are also probably more functionally heterogenous than we currently understand. Therefore, further investigations into the within mycorrhizal type functional heterogeneity and effects on soil microbiomes and functions is needed, which is beyond the scope and aim of this study.

Reviewer #3 (Remarks to the Author):

The manuscript has much improved and I really like it now. Earlier I was not so convinced that it was strong enough for Nature Communications (in that form), but now it includes many appealing data and the role of dark septate endophytes is much clearer shown now. I have a few minor comments:

Response: We thank the reviewer for reviewing the revised manuscript, and appreciate the helpful and constructive comments.

Line 2: title suggestion: dark septate fungi drive tree root and soil microbiome composition and function in European forests.

- if you would choose this title (and the editor agrees), obviously it needs to be made clear in the discussion that further field and experimental studies are needed to verify the observations made here (especially the important role of dark septate endophytes).-

Response: Thanks for the suggestions, we have taken them into account, we have tried to keep our title as broad and appealing as possible without overstating the findings.

Abstract: it would be worthwhile to mention the effects of dark septate endophytes (DSE) on pathogens (e.g. linked to the results presented in figure 3). Generally, the effects of DSE could even be more strongly emphasized in the abstract, also specifically mentioning their effects on a range of parameters including effects on pathogens, on fungal diversity, on functional gene composition etc. (to “visualize” the strong effects to the reader).

Response: We appreciate the suggestions, we have taken them into account, and mentioned the broad effects of DSE in the abstract, at the same time we are careful not to overemphasize the effects of DSE in our study.

L97-101: very long sentence. Consider to have two sentences. In general, the manuscript could benefit a little (minor issue, no major issue) if a native speaker goes over it (this reviewer is also not a native speaker).

Response: Thanks for pointing this out, we have had the manuscript read by a native English speaker, and have closely gone through to improve the grammar and tried to make our sentences simpler and easier to follow.

The discussion on the importance of DSE and their role in the soil is very useful (including potential mechanistic insights of how DSE influence pathogens and microbiome composition).

Response: Thanks, we are glad that it is useful

L446: the description of DSE detection (microscopy) is very short. Please add a few pictures (supplement) to show how DSE look like and explain the morphological characteristics used to count a DSE hyphae in more detail.

Response: We have now extended our methods description of DSE colonization quantification (now L549-560) and added photos of DSE structures from our root samples in the supplementary information Fig. S10.

Reviewer #4 (Remarks to the Author):

The authors present a thorough contribution detailing the potential effects of EcM, AM, and DSE on soil and root microbiomes and functional potentials. While I was charged by the editor to address one specific aspect of this work, namely the efficacy of previous revisions in responding to previous reviewers, I do also include a few other concerns here.

Overall, the authors present interesting results.

Response: We thank the reviewer for the assessment of our manuscript and constructive comments.

Lines 48-50: I do not think that this is a fair statement. There have been many previous studies that have explored how root symbionts impacts below ground studies, even with differential mycorrhizal status. This statement suggests a poor understanding of literature from the last 20 years.

Response: We thank the reviewer for this comment. We removed this statement from the manuscript as we do not think it is directly related to our main questions and hypotheses. We also note that we proceed to discuss existing and growing evidence of how root symbionts impact belowground properties/processes (L48-59).

Lines 105-106: Why do you think that EcM does not correlated with EcM here? And do you think that this lack of correlation where there should be one is concerning? How does this reflect of the rest of the data analyses?

Response: We note that this relationship is now significant after considering the random effect of tree species together with site/plot. The lack of a pattern is driven by *Alnus glutinosa*, when it is removed from the analysis the correlation is stronger (*Betula* and *A. incana*: $R^2m = 0.36$). We are unsure why this is the case as all EcM colonization analysis was performed by the same person using the same method, that being said it would be helpful for further studies to compare molecular quantification with microscopy quantification of colonization across a larger number of species, to determine their relationship to one another better so that EcM relative abundances could be used as a proxy for root colonization analysis, also such results might be improved by standardizing relative abundances based on qPCR of ITS copies to get a more absolute abundance measure that might correlate more strongly with microscopy colonization quantification.

Lines 168-178: DSE can be hard to find and can be prone to false negatives. That is, if you have a zero for colonization, how confident are you that it is a true zero? But these zeros are included in all of these analyses. It would be nice to omit these zeros and re-run these statistics to confirm that these general patterns still hold without a plethora of zero counts potentially artificially pulling the slope down,

Response: We thank the reviewer for this comment. We spent 1.5 hours per sample carefully analyzing each root sample to quantify DSE and AM colonization so we are pretty confident that our colonization percentages are accurate for the segments of roots we analyzed. Furthermore, we note that we present and analyze data on mean colonization values from 5 different trees per species per site. As shown in the figures and also the data values listed in the supplementary material, there is in fact only a single data point that is a true zero, meaning that on 5 replicate trees of that species at that site no DSE structures were observed, while there is a number of other data points that are very close to zero, but not absolute zeros. That being said, we have re-run the models for the relationships we present for DSE colonization with various microbiome properties in Fig. 3 and 4, we removed all mean DSE colonization values less than 2 % (6 data points), then ran the LME models again. We provide here a comparison of the results between data with removed low DSE values vs the complete dataset:

Relative abundance of plant pathogens (soil)

With all data ($R^2m = 0.29$, $p = 9e-04$), with removed data ($R^2m = 0.28$, $p = 9e-05$)

Ratio of bacteria/fungi (soil)

With all data ($R^2m = 0.18$, $p = 2e-03$), with removed data ($R^2m = 0.15$, $p = 0.01$)

Bacterial CAZyme diversity (soil)

With all data ($R^2m = 0.21$, $p = 7e-04$), with removed data ($R^2m = 0.21$, $p = 2e-03$)

Bacterial functional gene diversity (roots)

With all data ($R^2m = 0.36$, $p = 2e-05$), with removed data ($R^2m = 0.26$, $p = 1e-03$)

N cycling gene diversity (roots)

With all data ($R^2m = 0.37$, $p = 6e-05$), with removed data ($R^2m = 0.31$, $p = 1e-03$)

Based on this, we conclude that removing the low values does not substantially change the results and the general patterns remain, therefore we use the full range of DSE colonization results in our analysis reported in the manuscript.

180-185: I would suggest omitting this (and in methods) as it adds little to the story.

Response: We agree, now omitted

My biggest concern is that plant species was not controlled for. You have the categories of AM, EcM and DSE across a few species within genera. But individual species are not controlled for in analyses. With the exception of a recent paper in Fagaceae, almost all literature suggests that species (really, host genetics) play the predominant role in structuring mycorrhizal and soil communities. It is possible that all patterns seen here are species effects, rather than symbiont effects. This must be accounted for and discussed.

Response: We thank the reviewer for pointing out this. To address this, we have now re-run all analysis to take into account the effect of tree species. We have re-run linear mixed-effects models (including in piecewise SEM) to include host identity as part of a crossed random effect structure with plot embedded in site e.g., + (1| Site:Plot) + (1| Tree_species), whereas previously we only had Site/plot as a random effect. We have also re-run all PERMANOVA on community and gene compositions now using the PERMANOVA+ routine from the software Primer v7, which allows for the specification of random effects structures, and we present the new results in the manuscript (Fig. 5). The new analysis has slightly changed the results but did not affect our major conclusions. We have now changed and reshuffled the figures to reflect only relationships that are robust to the effects of site and tree species. Site was generally more important than host identity, and host identity often explained very little variation; we include all model outputs in the supplementary material.

REVIEWERS' COMMENTS

Reviewer #1 (Remarks to the Author):

The authors have satisfied my concerns.

Reviewer #4 (Remarks to the Author):

This MS is greatly improved and I commend the authors' for taking previous reviews to heart. This is a very extensive study and an interesting one.

Overall, this is a great study and I just have a few points. 1) All of the major results are correlation-based, which is to be expected for this type of work. But, the authors' overstate the level of inference possible based on these correlations. With correlational analyses, one cannot really understand mechanisms so all discussions need to take this into account and not over-promise what these data can and cannot tell us. 2) Lines 211-224 should also include some measure of effect (R-squared or similar) rather than just P-values. 3) Be very careful with discussing DSE colonization and genes, specifically TEs. The authors did not sequence metagenomes particularly deep, thus I can presume that there were not very many high-length MAGs, which would be needed to tie these TEs to taxa, they make a big leap trying to connect these to DSE.

Reviewer #1 (Remarks to the Author):

The authors have satisfied my concerns.

Response: We thank the reviewer for going through our revised manuscript, and we thank them for their constructive reviews of previously submitted versions that have greatly improved the manuscript.

Reviewer #4 (Remarks to the Author):

This MS is greatly improved and I commend the authors' for taking previous reviews to heart. This is a very extensive study and an interesting one.

Overall, this is a great study and I just have a few points. 1) All of the major results are correlation-based, which is to be expected for this type of work. But, the authors' overstate the level of inference possible based on these correlations. With correlational analyses, one cannot really understand mechanisms so all discussions need to take this into account and not over-promise what these data can and cannot tell us. 2) Lines 211-224 should also include some measure of effect (R-squared or similar) rather than just P-values. 3) Be very careful with discussing DSE colonization and genes, specifically TEs. The authors did not sequence metagenomes particularly deep, thus I can presume that there was not very many high-length MAGs, which would be needed to tie these TEs to taxa, they make a big leap trying to connect these to DSE.

We thank the reviewer for their time and for going through the manuscript again and for helping to improve the overall quality of the manuscript. In regards to point #1 we have tempered our discussion and the manuscript overall by mentioning 'potential' effects and associations due to the correlation-based nature of our analysis. In regards to point #2 we have now listed the explained variance (%) and pseudo-F values to accompany the p-values (now on Lines 215-235). In regards to point #3 the reviewer is correct that we did not sequence very deep and did not construct MAGs, so we cannot tie TEs to taxa other than broadly to kingdoms. To address this concern we have now removed our statement from the discussion linking DSE to TEs in our study.